# What is a Good Metric to Study Generalization of Minimax Learners?

**Asuman Ozdaglar**[†]   **Sarath Pattathil**[†]   **Jiawei Zhang**[†*]   **Kaiqing Zhang**[†‡]
[†]Massachusetts Institute of Technology   [‡]University of Maryland, College Park
{asuman,sarathp,jwzhang,kaiqing}@mit.edu

## Abstract

Minimax optimization has served as the backbone of many machine learning problems. Although the *convergence behavior* of optimization algorithms has been extensively studied in minimax settings, their *generalization* guarantees in the stochastic setting, i.e., how the solution trained on empirical data performs on the unseen testing data, have been relatively underexplored. A fundamental question remains elusive: *What is a good metric to study generalization of minimax learners?* In this paper, we aim to answer this question by first showing that *primal risk*, a universal metric to study generalization in minimization, fails in simple examples of minimax problems. Furthermore, another popular metric, the *primal-dual risk*, also fails to characterize the generalization behavior for minimax problems with nonconvexity, due to non-existence of saddle points. We thus propose a new metric to study generalization of minimax learners: the *primal gap*, to circumvent these issues. Next, we derive generalization bounds for the primal gap in nonconvex-concave settings. As byproducts of our analysis, we also solve two open questions: establishing generalization bounds for primal risk and primal-dual risk in the strong sense, i.e., without strong concavity or assuming that the maximization and expectation can be interchanged, while either of these assumptions was needed in the literature. Finally, we leverage this new metric to compare the generalization behavior of two popular algorithms – gradient descent-ascent (GDA) and gradient descent-max (GDMax) in stochastic minimax optimization.

## 1   Introduction

Stochastic minimax optimization, a classical and fundamental problem in operations research and game theory, involves solving the following problem:

$$\min_{w \in W} \max_{\theta \in \Theta} E_{z \sim P_z}[f(w, \theta; z)].$$

More recently, such a minimax formulation has received increasing attention in machine learning, with significant applications in generative adversarial networks (GANs) [Goodfellow et al., 2014], adversarial learning [Madry et al., 2017], and reinforcement learning [Chen and Wang, 2016, Dai et al., 2018]. Most existing works have focused on the *optimization* aspect of the problem, i.e., studying the rates of convergence, robustness, and optimality of algorithms for solving an empirical version of the problem where it approximates the expectation by an average over a sampled dataset, in various minimax settings including convex-concave [Nemirovski et al., 2009, Monteiro and Svaiter, 2010], nonconvex-concave [Lin et al., 2020, Rafique et al., 2018], and certain special nonconvex-nonconcave [Nouiehed et al., 2019, Yang et al., 2020] problems.

However, the optimization aspect is not sufficient to achieve the success of stochastic minimax optimization in machine learning. In particular, as in classical supervised learning, which is usually

---

[*]Corresponding Author.

36th Conference on Neural Information Processing Systems (NeurIPS 2022).

studied as a *minimization* problem [Hastie et al., 2009], the out-of-sample *generalization* performance is a key metric for evaluating the learned models. The study of generalization guarantees in minimax optimization (and related machine learning problems) has not received significant attention until recently [Arora et al., 2017, Feizi et al., 2020, Yin et al., 2019, Lei et al., 2021, Farnia and Ozdaglar, 2021, Zhang et al., 2021b]. Specifically, existing works along this line have investigated two types of generalization guarantees: *uniform* convergence generalization bounds, and *algorithm-dependent* generalization bounds. The former is more general and irrespective of the optimization algorithms being used, while the latter is usually finer and really explains what happens in practice, when optimization algorithms play an indispensable role. In fact, the former might not be able to explain generalization performance in deep learning, e.g., these bounds can increase with the training dataset size and easily become vacuous in practice [Nagarajan and Kolter, 2019], making the latter a more favorable metric for understanding the success of minimax optimization in machine learning.

Algorithm-dependent generalization for minimax optimization has been studied recently in [Farnia and Ozdaglar, 2021, Lei et al., 2021, Xing et al., 2021, Yang et al., 2022]. These papers build on the algorithmic stability framework developed in [Bousquet and Elisseeff, 2002], which are further investigated in [Hardt et al., 2016]. In particular, these works have studied *primal risk* and/or (variants of) *primal-dual risk* under different convexity and smoothness assumptions of the objective. Primal risk (see formal definition in §2) is a natural extension of the definition of risk from minimization problems. Primal-dual risk, on the other hand, is defined similarly but based on the duality gap of the solution. It is know that it is well-defined and can be optimized to zero only when the global saddle-point exists (i.e., min and max can be interchanged). Based on these metrics, [Farnia and Ozdaglar, 2021, Lei et al., 2021] compare the performance of specific algorithms, e.g., gradient descent-ascent (GDA) and gradient descent-max (GDMax). We provide a more thorough literature review in Section A, and a detailed comparison in Table A.

Although these metrics are natural extensions of generalization metrics from the *minimization* setting, they might not be the most suitable ones for studying generalization in stochastic *minimax* optimization, especially in the *nonconvex* settings that is pervasive in machine/deep learning applications, where the global saddle-point might not exist. In particular, we are interested in the following fundamental question:

*What is a good metric to study generalization of minimax learners[2]?*

In this paper, we make an initial attempt to answering this question, by identifying the inadequacies of the existing metric, and proposing a new metric, the *primal gap* that overcomes these inadequacies. We then provide generalization error bounds for our new metric, and discuss how it captures information not included in the other existing metrics, as well as discussing the tightness of the bounds. We summarize our contributions as follows.

**Contributions.** First, we introduce an example through which we identify the inadequacies of *primal risk*, a well-studied metric for generalization in stochastic minimax optimization, in capturing the generalization behavior of *nonconvex-concave* minimax problems. Second, to address the issue, we propose a new metric – the *primal gap*, which provably avoids the issue in the example, and derive its generalization error bounds. Next, we leverage this new metric to compare the generalization behavior of GDA and GDMax, two popular algorithms for minimax optimization and GAN training, and answer the question of *when does GDA generalize better than GDMax?* Moreover, we also address two open questions in the literature: establishing generalization error bounds for primal risk and primal-dual risk without strong concavity or assuming that the maximization and expectation can be interchanged, while at least one of these assumptions was needed in the literature [Farnia and Ozdaglar, 2021, Lei et al., 2021, Xing et al., 2021, Yang et al., 2022, Xiao et al., 2022]. Finally, under certain assumptions of the max learner, our results also generalize to the nonconvex-nonconcave setting.

## 2 Preliminaries

### 2.1 Problem formulation

In this paper, we consider the following (stochastic) minimax problem:

$$\min_{w \in W} \max_{\theta \in \Theta} E_{z \sim P_z} f(w, \theta; z). \tag{1}$$

---

[2]Hereafter, we use *learner* and *learning algorithm* interchangeably.

We make the following assumption on the sets $W$ and $\Theta$ throughout the paper.

**Assumption 1.** *$W$ and $\Theta$ are convex, closed sets, and we further assume that $W$ is compact with $\|w\| \leq M(W), \forall w \in W$. Here $M(W)$ is a constant dependent on the set $W$.*

Let $r(w, \theta) = E_{z \sim P_z} f(w, \theta; z)$. For a training dataset $S = \{z_1, \cdots, z_n\}$ with $n$ i.i.d. variables drawn from $P_z$, we define $r_S(w, \theta) = \frac{1}{n} \sum_{i=1}^{n} f(w, \theta; z_i)$. Next, we define the following quantity:

**Definition 1** (Primal risk (empirical/population))**.** ***Primal population risk** is given by*[3] *$r(w) = \max_{\theta \in \Theta} E_{z \sim P_z} f(w, \theta; z)$, and the **primal empirical risk** is given by: $r_S(w) = \max_{\theta \in \Theta} \frac{1}{n} \sum_{i=1}^{n} f(w, \theta; z_i)$.*

Throughout this paper, we use $(w_S, \theta_S)$ to denote a solution of the minimax problem: $\min_{w \in W} \max_{\theta \in \Theta} r_S(w, \theta)$. Notice that $(w_S, \theta_S)$ need not be a global saddle-point of $r_S$. Furthermore, we use $(w^*, \theta^*)$ to denote a solution of $\min_{w \in W} \max_{\theta \in \Theta} r(w, \theta)$. Once again, notice that $(w^*, \theta^*)$ may not be a saddle point of $r$.

The goal in Problem (1) is to minimize the primal population risk $r(w)$. Note that this function can be decomposed as

$$r(w) = r_S(w) + (r(w) - r_S(w)). \tag{2}$$

In practice, we only have access to $r_S(w, \theta)$, and our goal is to design algorithms for minimizing $r(w)$ using dataset $S$. Suppose $A$ is a learning algorithm initialized at $(w, \theta) = (0, 0)$. We define $(w_S^A, \theta_S^A)$ to be the output of Algorithm $A$ using dataset $S$.

From Equation (2), it is clear if we ensure $r_S(w_S^A)$ as well as $r(w_S^A) - r_S(w_S^A)$ are small, this would guarantee that $r(w_S^A)$ is small, which is the goal of Problem (1). Note that we can always ensure that $r_S(w_S^A)$ is small by using a good optimization Algorithm $A$ (if the problem is tractable). The main goal in the study of generalization is therefore to estimate the generalization error of the primal risk, as defined below.

**Definition 2.** *The generalization error for the primal risk is defined as: $\zeta_{gen}^P(A) = E_S E_A[r(w_S^A) - r_S(w_S^A)]$. Here the expectations are taken over the randomness in the dataset S, as well as any randomness used in the Algorithm A.*

This metric has been used to study generalization in stochastic minimization problems, i.e., when the maximization set $\Theta$ is a singleton, as well as several recent works in stochastic minimax optimization (see [Hardt et al., 2016, Farnia and Ozdaglar, 2021, Lei et al., 2021]).

We are interested in the question of when the solution to the empirical problem $w_S^A$ has good *generalization behavior*, i.e., when $E[r(w_S^A) - \min_{w \in W} r(w)]$ is small – $w_S^A$ is an approximate minimizer of the primal population risk $r$. In the next subsection, we briefly describe why the generalization error of the primal risk $\zeta_{gen}^P(A)$ is a good measure to study the generalization behavior in minimization problems.

### 2.1.1 $\quad \zeta_{gen}^P(A)$ for minimization problems

Consider a stochastic optimization problem of the form

$$\min_{w \in W} E_{z \sim P_z}[g(w; z)]. \tag{3}$$

We define the (minimization) primal risk (population and empirical version respectively) as: $r(w) = E_{z \sim P_z} g(w; z)$, and $r_S(w) = \frac{1}{n} \sum_{i=1}^{n} g(w; z_i)$. The generalization error $\zeta_{gen}^{P,min}(A)$ for the (minimization) primal risk is the same as in Definition 2 using the (minimization) primal risk.

Assume that the generalization error of the primal risk for an Algorithm $A$ is small, say $\zeta_{gen}^{P,min}(A) \leq \epsilon$. This implies that (from Definition 2): $E[r(w_S^A)] \leq E[r_S(w_S^A)] + \epsilon$. Note that the expectation is with respect to $S$ and $A$. Now, in order to show that $w_S^A$ has good generalization behavior, we first see that:

$$E[r(w_S^A) - \min_{w \in W} r(w)] \leq E[r_S(w_S^A)] + \epsilon - \min_{w \in W} r(w). \tag{4}$$

---

[3]Note that we slightly abuse the notation here by allowing $r$ and $r_S$ to have inputs that can be both $w$ and $(w, \theta)$. The distinction will be clear from context.

However, note that for minimization problems, since $E[r_S] = r$, we have that[4] $\min_{w \in W} r(w) \geq E[\min_{w \in W} r_S(w)]$, which gives us:

$$E[r(w_S^A) - \min_{w \in W} r(w)] \leq E[r_S(w_S^A)] + \epsilon - E[\min_{w \in W} r_S(w)] = E[r_S(w_S^A) - \min_{w \in W} r_S(w)] + \epsilon = \epsilon.$$

Therefore, for minimization problems, if the generalization error for primal risk is small, the solution to the empirical risk minimization problem has good generalization behavior. Next, we highlight some results in the literature which discusses generalization error bounds of the primal risk. These results depend on the concept of algorithmic stability we use later.

## 2.2 Stability of algorithms

Stability analysis is a powerful tool to analyze the generalization behavior of algorithms (see [Bousquet and Elisseeff, 2002]). In this section, we will review some definitions and theoretical results about stability bounds existing in the current literature. More specifically, in this paper, we adopt the following definition of stability:

**Definition 3** ($\epsilon$-stable Algorithm). *Suppose that $A$ is a randomized algorithm for solving the stochastic minimax problem. We define $(w_S^A, \theta_S^A)$ as the output of Algorithm $A$ using dataset $S$. We say $S$ and $S'$ are neighboring dataset if they defer only in one sample. An Algorithm $A$ is defined to be $\epsilon$-stable if $E_A \|w_S^A - w_{S'}^A\| \leq \epsilon$ and $E_A \|\theta_S^A - \theta_{S'}^A\| \leq \epsilon$ for any neighboring datasets $S$ and $S'$.*

[Hardt et al., 2016] gives the following basic result for the generalization error of $r_S(w)$.

**Theorem 1** ([Hardt et al., 2016]). *Consider the (stochastic) minimization problem defined in 3. Suppose $g(\cdot; z)$ is $\bar{L}$-Lipschitz continuous, i.e., $\forall z$, it holds that $\|g(w_1; z) - g(w_2; z)\| \leq \bar{L} \|w_1 - w_2\|, \forall w_1, w_2 \in W$. Then, for an $\epsilon$-stable Algorithm $A$, we have $|E_S E_A[r(w_S^A) - r_S(w_S^A)]| \leq \bar{L}\epsilon$.*

Unfortunately, Theorem 1 cannot be directly extended to analyze the generalization behavior of minimax learners, because we have an additional maximization step before taking expectation. Under certain additional conditions, primal risk can be a valid metric for minimax learners. We provide theoretical justifications in Section B, instantiated by an adversarial training example. We deal with the more general cases next.

# 3 Primal Gap: A New Metric to Study Generalization

The main idea which leads to the success of $\zeta_{gen}^P(A)$ as a metric to study generalization for minimization learners is that $E[\min_{w \in W} r_S(w)] \leq \min_{w \in W} r(w)$, which is no longer the case in the minimax case. In fact, in this section, we show that a good bound for the generalization error of primal risk does not imply good generalization behavior in minimax problems. We then propose a new metric, the primal gap, which circumvents these issues and provides insights into the generalization behavior in minimax problems.

## 3.1 Primal risk can fail for minimax learners

We provide an example where the generalization error of the primal risk is small, but the final solution to the empirical problem has poor generalization behavior. In this example, the minimizer of $r_S(w)$ is suboptimal for $r(w)$ with high probability, and $E_S[r(w_S) - r(w^*)]$ is large.

**Example 1** (Analytical example). *Let $y \sim N(0, 1/\sqrt{n})$ be a Gaussian random variable in $\mathbb{R}$. Define the truncated Gaussian variable $z \sim P_z$ as follows: $z = y$ if $|y| < \lambda \log n/\sqrt{n}$ and $z = \lambda \log n/\sqrt{n}$ if $y \geq \lambda \log n/\sqrt{n}$. Let $f(w, \theta; z) = \frac{1}{2}w^2 - \left(\frac{1}{2n^2}\theta^2 - z\theta + 1\right)w$, where $w \in W = [0, 1]$, $\theta \in \Theta = [-\lambda n, \lambda n]$ with a sufficiently large $\lambda > 0$, and $z_i \sim P_z$ be i.i.d truncated Gaussian variables. Then, we have $r_S(w, \theta) = \frac{1}{2}w^2 - \left(\frac{1}{2n^2}\theta^2 - \frac{\sum_{i=1}^n z_i}{n}\theta + 1\right)w$, and*

$$r(w, \theta) = \frac{1}{2}w^2 - \left(\frac{1}{2n^2}\theta^2 + 1\right)w. \tag{5}$$

*Note that this leads to the primal population risk function: $r(w) = \frac{1}{2}w^2 - w$.*

---

[4]Here we use the fact that $E_z[\min_x f(x, z)] \leq \min_x E_z[f(x, z)]$.

*It is not hard to see that we always have $r_S(w) \geq r(w)$. Note that this means $\zeta_{gen}^P(A) \leq 0$, and thus we have a small generalization error for primal risk. However, we can prove that for large enough $\lambda$,*

$$E_S[r(w_S) - r(w^*)] \geq 0.02. \tag{6}$$

*This means that $w_S$ has a constant error compared to $w^*$ in terms of the population risk, despite that its generalization error is small. This phenomenon is due to that $\min_{w \in W} r_S(w) - \min_{w \in W} r(w) > c$ for some $c > 0$, and hence minimizing $r_S(w)$ is very different from minimizing $r(w)$.*

This example shows that the generalization error of primal risk is not a good measure to study generalization in minimax learners. The main drawback is that $\min_w r_S(w)$ and $\min_w r(w)$ can be very different. We now introduce another more practical example, from GAN training, to further illustrate this point.

**Example 2** (GAN-training example). *Suppose that we have a real distribution $P_r$ in $\mathbb{R}^d$ which can be represented as $G^*(y)$ with $y \in \mathbb{R}^k$ drawn from a standard Gaussian distribution $P_0$ and a mapping $G^* : \mathbb{R}^k \to \mathbb{R}^d$. For an arbitrary generator $G$, we define $P_G$ to be the distribution of the random variable $G(y)$ with $y \sim P_0$. So our goal is to find a generator $G$ such that $P_G = P_r$. GAN is a popular tool for solving this problem. Consider a GAN with generator $G$, parametrized by $w$ and discriminator $D$ parametrized by $\theta$. The goal of GAN training is to find a pair of a generator $G$ and a discriminator $D$ that solves the minimax problem:*

$$\min_G \max_D \quad \{E_{x \sim P_r} \phi(D(x)) + E_{x \sim P_G}[\phi(1 - D(x))]\}$$
$$= \min_w \max_\theta \quad \{E_{x \sim P_r} \phi(D_\theta(x)) + E_{y \sim P_0}[\phi(1 - D_\theta(G_w(y)))]\},$$

*where $\phi : \mathbb{R} \to \mathbb{R}$ is concave, monotonically increasing and $\phi(u) = -\infty$ for $u \leq 0$. To connect to the minimax formulation in (1), we note that $z = (x, y)$, and $P_z = P_r \times P_0$. Also, we denote*

$$r(w, \theta) = E_{x \sim P_r} \phi(D_\theta(x)) + E_{y \sim P_0}[\phi(1 - D_\theta(G_w(y)))]$$

*to be the population risk. We now give the empirical version of this problem. Let $S_1 = \{x_1, \cdots, x_n\}$ and $S_2 = \{y_1, \cdots, y_n\}$. Let $S = S_1 \cup S_2$ and $r_S(w, \theta) = \frac{1}{n}(\sum_{i=1}^n \phi(D_\theta(x_i) + \phi(1 - D_\theta(G_w(y_i))))$. We assume that $P_{G_w}$ has the same support set as $P_r$. Moreover, we assume that $\|w - w^*\| \leq 0.5$ and $G_w(y)$ is 1-Lipschitz w.r.t. $w$ for any $y$. Here $w^*$ denotes the parameter for which $G_{w^*} = G^*$. Then, combining Theorem B.1 in [Arora et al., 2017] and the Lipschitz continuity of $G_w(y)$ as well as $\|w - w^*\| \leq 0.5$, we have that the distance between the sets $S_1$ and $\{G_w(y_1), G_w(y_2), \cdots, G_w(y_n)\}$ will be larger than $0.6$ with probability greater than $1 - O(n^2/e^d)$. Now, if $n$ is only of polynomial size of $d$, the optimal discriminator for disjoint datasets outputs $1$ on one dataset, and $0$ on the other. On the other hand, when $w = w^*$, the optimal discriminator for the population problem outputs $1/2$ for any sample it receives. Combining these two results, we have: $E_S[\min_{w \in W} r_S(w) - \min_{w \in W} r(w)] \geq (1 - \delta)(2\phi(1) - 2\phi(1/2))$, which is bounded away from $0$.*

Note that in this example, we also have $E_S[\min_w r_S(w) - \min_w r(w)] > 0$, implying that using $\zeta_{gen}^P(A)$ might not be a good way to characterize the generalization behavior in GAN training. To address this issue, we next define a new metric, the primal gap, and use its generalization error to study the generalization of minimax learners.

## 3.2 Primal gap to the rescue

The population and empirical versions of the primal gap are defined as follows:

**Definition 4** (Primal gap (empirical/population)). *The **population primal gap** is defined as $\Delta(w) = r(w) - \min_{w \in W} r(w)$, and the **empirical primal gap** is defined as $\Delta_S(w) = r_S(w) - \min_{w \in W} r_S(w)$.*

Notice that these two primal gaps can always take $0$ at $w_S \in \arg\min_{w \in W} r_S(w)$ and $w^* \in \arg\min_{w \in W} r(w)$ respectively even if the saddle point of problem (2) does not exist. Next, we define the expected generalization error of this primal gap as follows:

**Definition 5.** *The generalization error for the primal gap is $\zeta_{gen}^{PG}(A) = E_S E_A[\Delta(w_S^A) - \Delta_S(w_S^A)]$.*

Notice the fact that

$$E_S E_A[r(w_S^A) - \min_{w \in W} r(w)] = E_S E_A[r_S(w_S^A) - \min_{w \in W} r_S(w)] + \zeta_{gen}^{PG}(A).$$

If $E_S E_A[r_S(w_S^A) - \min_{w \in W} r_S(w)]$ is small and $\zeta_{gen}^{PG}(A)$ is small (or large), then $E_S E_A[r(w_S^A) - \min_{w \in W} r(w)]$ is small (or large).

Now we provide bounds on $\zeta_{gen}^{PG}(A)$ for a stable algorithm $A$, and show that in Example 1, $\zeta_{gen}^{PG}(A)$ cannot be small (unlike $\zeta_{gen}^{P}(A)$).

### 3.3 Relationship between generalization and stability

We provide bounds for the generalization error of the primal gap (Definition 5) for an $\epsilon$-stable Algorithm $A$. We will focus on the nonconvex-concave case where the following assumptions are made throughout the rest of the paper.

**Assumption 2.** *The function $f$ in Problem (1) is nonconvex-concave, i.e., $f(w, \cdot; z)$ is a concave function for all $w \in W$ and for all $z$.*

Next we define the notion of *capacity*, which will play a key role in the bounds we derive for $\zeta_{gen}^{PG}(A)$.

**Definition 6** (Capacity). *For any $w \in W$ and any constraint set $\Theta$, we define*

$$\Theta(w) = \arg\max_{\theta \in \Theta} r(w, \theta) \qquad \Theta_S(w) = \arg\max_{\theta \in \Theta} r_S(w, \theta).$$

*We define the capacities $C_p$ and $C_e$ as:*

$$C_p(\Theta) = \max_{w \in W} \text{dist}(0, \Theta(w)), \qquad C_e(\Theta) = \max_{S} \max_{w \in W} \text{dist}(0, \Theta_S(w)),$$

*where $\text{dist}(p, \mathcal{S})$ denotes the distance between a point $p$ to a set $\mathcal{S}$ in Euclidean space, i.e., $\text{dist}(p, \mathcal{S}) := \inf_{q \in \mathcal{S}} \|p - q\|_2$. For the constraint set in Problem (1), we succinctly denote the capacities as $C_p$ and $C_e$, respectively.*

The norm of the model parameter (its distance to $0$) is usually viewed as the metric for the complexity of the model. In fact, the norm of the optimal solution determines the Rademacher complexity of the function class in statistical learning theory [Vapnik, 1999]. Moreover, in deep learning, minimum-norm solution of overparameterized neural networks is well-known to enjoy better generalization performance [Zhang et al., 2021a]. Hence, we view the capacity constant $C_e$ and $C_p$ as natural metrics to capture the model complexity for the best response of the max learner, i.e., the power of the maximizer, when using the empirical data set and population data respectively.

Now, we are ready to discuss the relationship between the stability bound and the generalization error of algorithms in nonconvex-concave minimax problems. All proofs have been deferred to the appendix. We make the following assumptions throughout the paper:

**Assumption 3.** *The gradient of $f$ is $\ell$-Lipschitz-continuous for all $z$, i.e., for all $z$*

$$\|\nabla f(w_1, \theta_1; z) - \nabla f(w_2, \theta_2; z)\| \leq \ell(\|w_1 - w_2\| + \|\theta_1 - \theta_2\|), \quad \forall w_1, w_2 \in W, \quad \forall \theta_1, \theta_2 \in \Theta.$$

*Moreover, fixing $w \in W$, the partial gradient $\nabla_\theta f(w, \cdot; z)$ is $\ell_{\theta\theta}$-Lipschitz continuous with respect to $\theta$ for all $z$, i.e., $\|\nabla_\theta f(w, \theta_1; z) - \nabla_\theta f(w, \theta_2; z)\| \leq \ell_{\theta\theta}\|\theta_1 - \theta_2\|, \forall w \in W, \quad \forall \theta_1, \theta_2 \in \Theta.$*

**Assumption 4.** *For any $\Theta_1 \subseteq \Theta$, we assume that $f$ is $L(\Theta_1)$-Lipschitz-continuous with respect to $w \in W, \theta \in \Theta_1$ for all $z$, i.e., $\|f(w_1, \theta_1; z) - f(w_2, \theta_2; z)\| \leq L(\Theta_1)(\|w_1 - w_2\| + \|\theta_1 - \theta_2\|), \quad \forall w_1, w_2 \in W, \forall \theta_1, \theta_2 \in \Theta_1$, and the gradient $\nabla f(w, \theta; z)$ is uniformly bounded as $\|\nabla_{w,\theta} f(w, \theta; z)\| \leq L(\Theta_1)$ for all $z$ and $w \in W, \theta \in \Theta_1$. Moreover, $f(w^*, \cdot; z)$ is $L_\theta^*$-Lipschitz continuous with respect to $\theta$ where $w^* \in \arg\min_{w \in W} r(w)$. We also define $L := L(B(0, 2C_p + 1) \cap \Theta)$ and $L_r := L(B(0, r) \cap \Theta)$, where $B(v, r)$ denotes the $l_2$-ball with radius $r$ centered at $v$.*

Note that we can decompose the generalization error of the primal gap as follows:

$$\zeta_{gen}^{PG}(A) := E_S E_A[\Delta(w_S^A) - \Delta_S(w_S^A)] = \zeta_{gen}^{P}(A) + E_S\Big[\min_{w \in W} r_S(w) - \min_{w \in W} r(w)\Big].$$

We now provide a generalization error bound for the primal risk $\zeta_{gen}^{P}(A)$. To the best of our knowledge, it is the first bound for $\zeta_{gen}^{P}(A)$ in the nonconvex-concave (not strongly concave) setting.

**Lemma 1.** *The generalization error of the primal risk of an $\epsilon$-stable Algorithm $A$ for a minimax problem with concave maximization problem can be bounded by $\zeta_{gen}^P(A) \leq \sqrt{4L\ell C_p^2} \cdot \sqrt{\epsilon} + \epsilon L$.*

We show that this dependence on $\epsilon$ is *tight* in Section D.2. Since we already have the generalization error for the primal risk $E_S E_A[r(w_S^A) - r_S(w_S^A)]$ from Lemma 1, we only need to estimate

$$E_S E_A\big[\min_{w \in W} r_S(w) - \min_{w \in W} r(w)\big] = E_S\big[\min_{w \in W} r_S(w) - \min_{w \in W} r(w)\big] \qquad \text{[Primal Min Error].} \quad (7)$$

The following theorem gives the generalization bound of the primal gap using the upper bound from Lemma 1 and bounding the Primal Min Error in Equation (7).

**Theorem 2.** *Suppose Algorithm $A$ is $\epsilon$-stable. The generalization error bound of the primal gap is given by $\zeta_{gen}^{PG}(A) \leq \sqrt{4L\ell C_p^2} \cdot \sqrt{\epsilon} + \epsilon L + 4L_\theta^* C_e/\sqrt{n}$.*

The first term in the bound above is from the generalization bound of the primal risk, as shown in Lemma 1. Note that the bound in Lemma 1 only involves $C_p$, as the key in the analysis is to upper-bound the population risk $r(w_S^A)$, which requires bounding the power of the maximizer using the population capacity $C_p$. This reflects the intuition that the power of the maximizer should affect the generalization behavior of minimax learners, and the stronger the maximizer is, the harder for the learner to generalize. On the other hand, the bound in Theorem 2 additionally involve $C_e$, the empirical capacity. Technically, $C_e$ (instead of $C_p$) appears since we need to bound $\min_w r_S(w)$ (defined on the empirical dataset) in the Primal Min Error term in (7). We show the *tightness* of this bound in Section D.4. The appearance of $C_e$ reflects the intuition that the difference between the maximizers of the empirical and population risks should make a difference in characterizing the generalization of minimax learners. This intuition cannot be captured by the generalization error of the primal risk, as in Lemma 1. Note that in the minimization case, the Primal Min Error can be upper-bounded directly by zero, and such a distinction disappears, making primal risk a valid metric.

### 3.4 Revisiting Example 1

In Example 1, we have that the primal risk generalizes well, but the solution $w_S$ does not, thereby indicating that the generalization error of the primal risk may not be a good metric to study generalization behavior of minimax problem. Note that in Example 1, we have $\zeta_{gen}^{PG}(A) \geq 0.02$ while $\zeta_{gen}^P(A) \leq 0$. We already know that this algorithm can not generalize well. Therefore, $\zeta_{gen}^{PG}(A)$ can capture the generalization behavior better than $\zeta_{gen}^P(A)$. This shows that the primal min error $\zeta_{gen}^{PM}$ also plays an important role in analysing the generalization for minimax problems. As shown in the appendix (Proposition 4), we have

$$E_S[\min_{w \in W} r_S(w) - \min_{w \in W} r(w)] \geq 0.005. \qquad (8)$$

On the other hand, it is easy to compute that $L_\theta^* = \lambda \log n/\sqrt{n}$ and $C_e = \lambda n \log n$. Therefore, by Theorem 2, we have an upper bound for the Primal Min Error (see (7)): $E_S[\min_{w \in W} r_S(w) - \min_{w \in W} r(w)] \leq 4L_\theta^* C_e/\sqrt{n} = 4 \log n$, which is tight up to a $\log$ factor according to (8). Therefore, the primal gap has a constant generalization error which is consistent with the observation that the solution to the empirical problem does not have good generalization behavior.

### 3.5 Nonconvex-nonconcave case

In this section, we extend our results to the nonconvex-nonconcave setting. We will show that under certain assumptions on the inner maximization problem, we can derive generalization error bounds for the primal risk and primal gap in terms of algorithmic stability. We make the following assumptions on the inner maximization problem:

**Assumption 5.** *For any $\gamma > 0$, there exists an algorithm which outputs $\theta_P^\gamma(w)$, for the inner maximization problem $\max_{\theta \in \Theta} r(w, \theta)$, satisfying the following conditions: 1) $r(w) - r(w, \theta_P^\gamma(w)) \leq \gamma$; 2) $\|\theta_P^\gamma(w) - \theta_P^\gamma(w')\| \leq \frac{\lambda_p}{\gamma}\|w - w'\|$ with some constant $\lambda_p > 0$ for all $w, w' \in W$.*

**Assumption 6.** *For any $\gamma > 0$, there exists an algorithm which outputs $\theta_E^\gamma(S)$, for the inner maximization problem $\max_{\theta \in \Theta} r_S(w^*, \theta)$, satisfying the following conditions: 1) $r_S(w^*) - r_S(w^*, \theta_E^\gamma(S)) \leq \gamma$. 2) For any neighboring dataset $S$ and $S'$, $\|\theta_E^\gamma(S) - \theta_E^\gamma(S')\| \leq \frac{\lambda_e}{n\gamma}$ for some $\lambda_e > 0$.*

The following lemma gives sufficient conditions for these two assumptions to hold.

**Lemma 2.** *Consider constants $D_e \geq \gamma$ and $D_p \geq \gamma$.*

1. *Suppose gradient ascent with diminishing stepsizes $c_0/t$ for the problem $\max_{\theta \in \Theta} r(w, \theta)$ has convergence rate $r(w) - r(w, \theta^s) \leq D_p/s$. Then we define $\theta_p^\gamma(w)$ by performing $s = D_p/\gamma$ steps of gradient ascent. Then, $\theta_p^\gamma(w)$ satisfies Assumption 5.*

2. *Suppose gradient ascent with constant stepsize $c_0$ for the problem $\max_{\theta \in \Theta} r(w, \theta)$ has convergence rate $r(w) - r(w, \theta^s) \leq D_p \eta^s$ for some constant $0 < \eta < 1$. Then we define $\theta_p^\gamma(w)$ by $s = \log(D_p/\gamma)/\log(1/\eta)$ steps of gradient ascent. Then, $\theta_p^\gamma(w)$ satisfies Assumption 5.*

3. *Suppose gradient ascent with diminishing stepsizes $c_0/t$ for the problem $\max_{\theta \in \Theta} r_S(w, \theta)$ has convergence rate $r_S(w) - r_S(w, \theta^s) \leq D_p/s$. Then we define $\theta_e^\gamma(S)$ by performing $s = D_e/\gamma$ steps of gradient ascent. Then, $\theta_e^\gamma(S)$ satisfies Assumption 6.*

4. *Suppose gradient ascent with constant stepsize $c_0$ for the problem $\max_{\theta \in \Theta} r_S(w, \theta)$ has convergence rate $r_S(w) - r_S(w, \theta^s) \leq D_e \eta^s$ for some constant $0 < \eta < 1$. Then we define $\theta_e^\gamma(w)$ by $s = \log(D_e/\gamma)/\log(1/\eta)$ steps of gradient ascent. Then, $\theta_e^\gamma(S)$ satisfies Assumption 6.*

**Remark 1.** *Note that for some practical nonconvex optimization problems in machine learning, gradient descent indeed converges to the global minima at a reasonably fast rate, e.g., in training deep overparametrized neural networks [Du et al., 2019], robust least squares problems [El Ghaoui and Lebret, 1997], phase retrieval and matrix completion [Ma et al., 2019]. Our Assumptions 5 and 6 can be viewed as an abstract summary of some benign properties of gradient descent for certain nonconvex optimization problems.*

Furthermore, we assume that $f(\cdot, \cdot; z)$ is $L$-Lipschitz[5] continuous in $W \times \Theta$. This, along with Assumptions 5 and 6, allows us to derive the generalization error bounds of the primal risk and primal gap in terms of algorithmic stability.

**Lemma 3.** *Suppose that Assumption 5 holds. If a minimax learner $A$ is an $\epsilon$-stable algorithm, we have $\zeta_{gen}^P(A) \leq L\epsilon + \sqrt{L\lambda_p}\sqrt{\epsilon}$.*

Similarly, we can derive the generalization bound for the primal gap given the above assumptions.

**Theorem 3.** *Suppose Assumptions 5 and 6 hold. Then we have $\zeta_{gen}^{PG}(A) \leq \zeta_{gen}^P(A) + \sqrt{L\lambda_e}/\sqrt{n}$.*

The proof of this theorem is similar to the proof of Lemma 3 and Theorem 2 and hence omitted.

## 4 Comparison of GDA and GDMax

In Section 3.3, we provide generalization bounds for the primal gap for any $\epsilon$-stable algorithm. In this section, we focus on two algorithms in particular – GDA and GDMax. These two algorithms are described in Algorithms 1 and 2 in Appendix E.

We note that though analyzing the *optimization* properties of GDA/stochastic GDA for solving the empirical minimax problem is an important topic, our focus in this paper is on studying the generalization behavior of these algorithms. We assume that the empirical version of the stochastic minimax problem can be solved by GDA and GDMax, i.e., we assume that GDA and GDMax satisfy the following assumption:

**Assumption 7.** *Let $A$ be a minimax learner, such as GDA or GDMax. Then we assume that $A$ has the following convergence rate: $E_A[r_S(w^t) - \min_{w \in W} r_S(w)] \leq (\phi_A(M(W)) + \phi_A(C_e))/\psi_A(t)$, where $M(W)$ is the maximum of the norms of $w$, and $\phi_A(s)$, $\psi_A(s)$ are nonnegative, increasing functions that tend to infinity as $s \to \infty$.*

For simplicity, throughout this section, we assume that $\|f(w, \theta; z)\| \leq 1$ for all $w, \theta$, and $z$. The next theorem provides a bound for the population primal gap $\Delta(w_S^A) := r(w_S^A) - \min_{w \in W} r(w)$. Note that the goal of any algorithm is to make this gap as small as possible.

For an Algorithm $A$ and subsets $W_0 \subseteq W, \Theta_0 \subseteq \Theta$, we define $A(W_0, \Theta_0)$ as the algorithm which restricts $A$ to solve (1) under constraint sets $W_0$ and $\Theta_0$. Specifically, $A(W, \Theta)$ is just $A$.

---

[5]Note that this is different from the $L$ defined for the nonconvex-concave case. Here $L$ captures the Lipschitz constant over the whole constraint set. In the nonconvex-concave case, $L = L(B(0, 2C_p + 1))$.

**Theorem 4.** *Let $w_S^{A,t}, \theta_S^{A,t}$ be the $t$-th iterate generated by Algorithm $A$ using dataset $S$. Assume that $\{\theta_S^{A,t}\} \subseteq \Theta_0 = \Theta_\theta^A$ for $t \leq T$ with probability $1 - \delta$ (due to the randomness in $S$) and $B(0, C_p) \subseteq \Theta_\theta^A$. Here $B(v, r)$ denotes the $l_2$-ball with radius $r$ centered at $v$. Let $A_0 = A(W, \Theta_0)$. Then after $T$ iterations of Algorithm $A$, the population primal gap can be bounded as:*

$$E_S[r(w_S^{A,T}) - \min_{w \in W} r(w)] \leq \underbrace{\zeta_{gen}^P(A_0)}_{I} + \underbrace{(\phi_{A_0}(M(W)) + \phi_{A_0}(C_e(\Theta_\theta^A)))/\psi_{A_0}(T) + 4L_\theta^* C_e(\Theta_\theta^A)/\sqrt{n}}_{II} + \delta,$$

*where $\zeta_{gen}^P(A_0) = E_S E_A[r(w_S^{A_0,T}) - r_S(w_S^{A_0,T})]$ is the generalization error of the primal risk of Algorithm $A_0$.*

**Remark 2.** *Theorem 4 builds a closer connection between generalization behavior and the dynamics of the minimax learner $A$. It shows that suitable restriction to the max learner can lead to better minimax learner, in terms of generalization. We make this clear in the comparison of GDA and GDMax by analyzing the three terms in Theorem 4.*

### 4.1 Analyzing the term $I$

First, we study the generalization error bound of the primal risk, i.e., $\zeta_{gen}^P$ in Theorem 4. For GDA, we can estimate $\zeta_{gen}^P$ by using Lemma 1. Therefore, it suffices to estimate the stability of GDA. We do this in the following lemma:

**Lemma 4.** *Let $c_0 = \max\{\alpha_0, \beta_0\}$, If we use diminishing stepsizes $\alpha_t = \alpha_0/t$ and $\beta_t = \beta_0/t$ for GDA for $T$ iterations, we have the stability bound $\epsilon^{GDA} \leq 2L_{\Theta_\theta^{GDA}} T^{c_0 \ell}/(n\ell)$.*

Now, since we have a bound for $\zeta_{gen}^P(A)$ for an $\epsilon$-stable Algorithm $A$ in Lemma 1, we can substitute the stability bound for GDA from Lemma 4 in this expression to get a bound on $\zeta_{gen}^P(GDA)$ for GDA. We do this in the next proposition. We can bound $\zeta_{gen}^P(A_0)$ for GDA by substituting the stability bound in Lemma 4 into Lemma 1 (letting $\epsilon = \epsilon^{GDA}$).

**Proposition 1.** *Let $c_0 = \max\{\alpha_0, \beta_0\}$ and assume that $f(\cdot, \cdot; z)$ is $L_{\Theta_\theta^{GDA}}$-Lipschitz-continuous inside the set $W \times \Theta_\theta^{GDA}$. For GDA with diminishing stepsizes $\alpha_0/t, \beta_0/t$ run for $T$ iterations (denoted by $GDA_T$), the generalization error of the primal risk can be bounded by:*

$$\zeta_{gen}^P(GDA_T) \leq (L_{\Theta_\theta^{GDA}})^{3/2} \sqrt{8C_p^2/\ell} \sqrt{T^{c_0 \ell}/n} + 2L_{\Theta_\theta^{GDA}}^2 T^{c_0 \ell}/(n\ell).$$

However, for GDMax, we can not compute a uniform stability bound that vanishes as $n$ goes to infinity. In fact, we can show from the following simple example that $\zeta_{gen}^P(\text{GDMax})$ can be a constant that is independent of $n$, which means that for the case where $r(w, \theta)$ is nonconvex-concave, the generalization error of primal risk of GDMax can be undesirable.

**Example 3** (Constant generalization error of primal risk for GDMax). *Consider a dataset $S$ with $n$ elements. Define the objective function: $f(w, \theta; z) = \left(\frac{w}{n^2} - z\right)\theta - \frac{\theta^2}{2n}$, where $w \in W = [-n\sqrt{n}, n\sqrt{n}], \theta \in \Theta = \mathbb{R}$ and $z$ is drawn from the uniform distribution over $\{-1/\sqrt{n}, 1/\sqrt{n}\}$. We have $r_S(w) = \frac{n^2}{2}\left(\frac{w}{n^2} - \frac{1}{n}\sum_{i=1}^n z_i\right)^2$, and $r(w) = \frac{w^2}{2n^2}$. Therefore, $\min_{w \in W} r(w) = 0$. From the definition of the function $f$ and the sets $W$ and $\Theta$, we have $\ell = 1/n^2, L = \mathcal{O}(1/\sqrt{n})$.*

*One step of GDMax can attain the minimizer of $r_S(w)$ (since it is a one dimensional quadratic problem), i.e., $w_S = n\sum_{i=1}^n z_i$ and $r_S(w_S) = 0$. Furthermore, we have $E_S r(w_S) = E[\frac{(\sum_{i=1}^n z_i)2}{2}] = 1/2 > 0$. Thus, $\zeta_{gen}^P(\text{GDMax}) = E[r(w_S) - r_S(w_S)] = 1/2 > 0$ cannot be made small.*

Therefore, from Proposition 1 and Example 3, we see that the bound for the expected population primal gap contains the term $\zeta_{gen}^P$ which cannot be bounded for GDMax, whereas can be bounded for GDA which leads us to the conclusion that GDA generalizes better than GDMax for such problems. However, it is possible to bound $\zeta_{gen}^P(\text{GDMax})$ in certain problems, and in this case the other terms in Theorem 4 become crucial. We analyze them next.

### 4.2 Analyzing the term $II$

As shown in Example 1, sometimes GDMax can have a good generalization bound for the primal risk. Therefore, we need to analyze the other two terms in Theorem 4, i.e., $(\phi_A(M_w) +$

$\phi_A(C_e(\Theta_\theta^A)))/\psi_A(T)$ and $L_\theta^* C_e(\Theta_\theta^A)/\sqrt{n}$. For these two terms, since $L_\theta^*$ is fixed, the constant $C_e(\Theta_\theta^A)$ is the key term which differentiates the performance of different algorithms.

By definition, the constant $C_e(\Theta_\theta^{GDMax})$ for GDMax is nearly $C_e$ (See Definition 6). Therefore, the population primal gap after $T$ steps of GDMax is dominated by $C_e$ if $C_e$ is large. However, the set $\Theta_\theta^{GDA}$ for GDA can be much smaller than $\Theta$, which implies that $C_e(\Theta_\theta^{GDA})$ can be much smaller than $C_e$. This phenomenon can be seen from Example 1: If we perform one step of GDMax with primal stepsize 1, we can attain $w^1 = w_S$. Then $E_S[r(w_S^1) - \min_{w \in W} r(w)] \geq 0.005$ from (6). For GDA, we can see that $w^1 = 1$ after one step of GDA with stepsize 1. Therefore, GDA generalizes better than GDMax. Generally, we have the following estimate of $C_e(\Theta_\theta^{GDA})$.

**Lemma 5.** *Let $L_0 = \max_z \|\nabla f(w_0, \theta_0; z)\|$. Let $c_0 = \max\{\alpha_0, \beta_0\}$. If we use diminishing stepsizes $\alpha_t = \alpha_0/t$ and $\beta_t = \beta_0/t$ for GDA, then after $T$ steps we have $\|\theta^t\| \leq T^{c_0 \ell} L_0/\ell$ for $t \in [T]$.*

Therefore, if $C_e$ is much larger than $C_p$, using GDA with $C_p \leq T^{c_0 \ell} L_0/\ell \leq C_e$ is better than GDMax. We make this more concrete in the context of GAN training next.

### 4.3 GAN training

We now study the specific case of GAN training to explore why GDA might generalize better than GDMax. This is numerically verified in the literature, such as Farnia and Ozdaglar [2021]. Specifically, we revisit Example 2, and consider a special case: $D$ is restricted to be a over-parametrized linear function with respect to $\theta$. Define the descriminator $D(x) = \Phi^T(x)v + b_0$, where $\Phi(x) = [\Phi_1(x), \cdots, \Phi_m(x)]^T \in \mathbb{R}^m$ is the feature matrix and $b_0 \in \mathbb{R}$. Also suppose that $G$ is parametrized by $w$ and $G^* = G_{w^*}$. Then the GAN problem can be written as $\min_{w \in W} \max_{\theta \in \Theta} r(w, \theta)$, where

$$r(w, \theta) = E_{x \sim P_r}[\phi(v^T \Phi(x) + b_0)] + E_{y \sim P_0}[\phi(1 - v^T \Phi(G_w(y)) - b_0)].$$

Here $\theta = (v, b_0)$. Assume that $\sqrt{\sigma_{\max}\left(E_{x \sim P_{G_w}} \Phi(x)\Phi^T(x)\right)} \leq \bar{\sigma}_{\max}/\sqrt{m}$, where $\sigma_{\max}(\cdot)$ denotes the largest singular value of a matrix and $\bar{\sigma}_{\max} > 0$ is a constant. Also assume that $E_{x \sim P_{G_w}} \Phi(x)\Phi^T(x)$ is full rank. Also, we assume that $|\phi'(\lambda)| \leq L_\phi$ for any $\lambda \in [0, 1]$. Therefore, we have $E[\|\nabla_\theta f(w, \theta; z)\|^2] \approx L_\phi^2 \bar{\sigma}_{\max}^2$. Then it is reasonable to assume that $\|\nabla f\| \leq \mathcal{O}(1)$.

**Lemma 6.** *Suppose $\Phi(x)$ is sub-Gaussian and the matrix*

$$Q_S = [\Phi(x_1) \quad \Phi(x_2) \cdots \quad \Phi(x_n) \quad \Phi(G_w(y_1)) \quad \cdots \quad \Phi(G_w(y_n))]$$

*is full column rank ($m > n$) with probability 1. Then with probability at least $1 - C\delta$ with some constant $C$, we have $\|\theta_S(w^*)\| \geq \Omega(\sqrt{n})$, where $\theta_S(w^*) \in \arg\max_{\theta \in \Theta} r_S(w^*, \theta)$.*

Now, for $\theta \in \arg\max_{\theta' \in \Theta} r(w^*, \theta')$, it can be easily seen that $v = 0, b_0 = 1/2$ in this case. Therefore, $C_p \approx 1/2$. Finally, combining the previous discussion on GDA in Lemma 5, and using the fact that $C_e$ is large from Lemma 6, we see from Theorem 4 that GDA can generalize better than GDMax. More detailed discussion of the GAN-training and Lemma 6 can be found in Section E.

## 5 Conclusions

In this paper, we first demonstrate the shortcomings of one popular metric, the primal risk, in terms of characterizing the generalization behavior of minimax learners. We then propose a new metric, the primal gap, whose generalization error overcomes these shortcomings and captures the generalization behavior of algorithms that solve stochastic minimax problems. Finally, we use this newly proposed metric to study the generalization behavior of two different algorithms – GDA and GDMax, and study cases where GDA has a better generalization behavior than GDMax. Future directions include further investigation of the proposed new metric, the primal gap, and deriving its (tighter) generalization error bounds in other structured stochastic minimax optimization problems in machine learning.

## Acknowledgments and Disclosure of Funding

Research was sponsored by the United States Air Force Research Laboratory and the United States Air Force Artificial Intelligence Accelerator and was accomplished under Cooperative Agreement Number

FA8750-19-2-1000. The views and conclusions contained in this document are those of the authors and should not be interpreted as representing the official policies, either expressed or implied, of the United States Air Force or the U.S. Government. The U.S. Government is authorized to reproduce and distribute reprints for Government purposes notwithstanding any copyright notation herein. S.P. acknowledges support from MathWorks Engineering Fellowship. A.O. and K.Z. were supported by MIT-DSTA grant 031017-00016. K.Z. also acknowledges support from Simons-Berkeley Research Fellowship.

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
