# Supplementary Materials for
# "What is a Good Metric to Study Generalization of Minimax Learners?"

## A    Related work

**Algorithms for minimax optimization.**    There is a vast literature on algorithms for minimax optimization. The most popular algorithms include the Extragradient (EG), the Optimistic Gradient Descent Ascent (OGDA) and the Gradient Descent Ascent and their variants. The EG algorithms introduced in [Korpelevich, 1976], has been analyzed in several papers including [Monteiro and Svaiter, 2010, Mokhtari et al., 2020a,b, Golowich et al., 2020b] for (strongly)convex-(strongly)concave problems. Another popular algorithm is OGDA introduced in [Popov, 1980] and has been analyzed in several recent works including [Daskalakis et al., 2017, Hsieh et al., 2019, Golowich et al., 2020a]. Once again, all these works focus on the (strongly)convex-(strongly)concave setting. Stochastic versions of these algorithms in similar settings have also been analyzed in several papers including [Nemirovski et al., 2009, Hsieh et al., 2019, Fallah et al., 2020]. A few papers including [Lin et al., 2020, Zhang et al., 2020, Huang et al., 2022, Zhang et al., 2021c, Ostrovskii et al., 2021b, Kong et al., 2019, Zhang et al., 2020] analyze gradient based algorithms in the nonconvex-(strongly)concave cases. Some papers including [Rafique et al., 2018, Yang et al., 2021, Ostrovskii et al., 2021a, Grimmer et al., 2020] analyze special cases of nonconvex-nonconcave (like nonconvex-PL) for algorithms like GDA and its variants. However, in this paper, we are interested in the generalization performance of these algorithms. We summarize below the most related literature that studies the generalization behavior in minimax optimization problems.

**Algorithm-independent generalization.**    Specific to the machine learning problems of GAN and adversarial training, there have been several papers studying the uniform convergence generalization bounds. [Arora et al., 2017] establish a uniform convergence generalization bound which depends on the number of discriminator parameters. [Wu et al., 2019] connect the stability-based theory to differential privacy ([Shalev-Shwartz et al., 2010]) in GANs and numerically study the generalization behavior in GANs. [Zhang et al., 2017, Bai et al., 2018] analyze the Rademacher complexity of the players to show the uniform convergence bounds for GANs. In the simpler Gaussian setting, [Feizi et al., 2020] and [Schmidt et al., 2018] derive bounds for GANs and adversarial training, respectively. The uniform convergence bounds for adversarial training have also been studied under several statistical learning frameworks, e.g., PAC-Bayes [Farnia et al., 2018], Rademacher complexity [Yin et al., 2019], margin-based [Wei and Ma, 2019], and VC analysis [Attias et al., 2019]. Recently, [Zhang et al., 2021b] investigate the generalization of empirical saddle point (ESP) solution in strongly-convex-concave problems using a stability-based approach. Note that these results are not specific to the optimization algorithms being used.

**Algorithm-dependent generalization.**    Algorithm specific generalization bounds for minimax optimization have attracted increasing attention. Based on the algorithmic stability framework in [Bousquet and Elisseeff, 2002], [Farnia and Ozdaglar, 2021] have established generalization bounds of standard gradient descent-ascent and proximal point algorithms under the convex-concave setting, and those of stochastic GDA and GDMax under the nonconvex-strongly concave setting. Concurrently, [Lei et al., 2021] derive high-probability generalization bounds for both convex-concave and weakly convex-weakly concave settings, with possibly nonsmooth objectives, also through the lens of algorithmic stability. Both works hinged on the metrics of *primal risk* and *primal-dual risk*. As shown in the present work, the former is not necessarily suitable to characterize the generalization behavior of minimax optimization, while the latter is known to be appropriate only when the saddle point exists, which is usually not the case in the nonconvex settings that are common in machine learning. Following this line of work, [Xing et al., 2021] provide generalization bounds specifically for adversarial training, which is essentially the primal risk, also using the algorithmic stability framework. Recently, [Yang et al., 2022] study the generalization of stochastic GDA under differential privacy constraints.

| Reference | Assumption | Metric | Rate |
|---|---|---|---|
| [Farnia and Ozdaglar, 2021] | NC-$\mu$-SC | PR | $L\sqrt{\kappa^2 + 1}\epsilon$ |
| [Lei et al., 2021] | NC-$\mu$-SC | PR | $L(1 + \kappa)\epsilon$ |
| [Lei et al., 2021] | $\mu$-SC-SC | PD | $\sqrt{2}L(1 + \kappa)\epsilon$ |
| This work (Theorem 2) | NC-C | PG | $\sqrt{4L\ell C_p^2} \cdot \sqrt{\epsilon} + \epsilon L + 4L_\theta^* C_e/\sqrt{n}$ |
| This work (Lemma 1) | NC-C | PR | $\sqrt{4L\ell C_p^2} \cdot \sqrt{\epsilon} + \epsilon L$ |
| This work (Theorem 7) | C-C | PD | $\left(\sqrt{4L\ell C_p^2} + \sqrt{4L\ell (C_p^w)^2}\right)\sqrt{\epsilon} + 2\epsilon L$ |

Table 1: Generalization bounds for $\epsilon$-stable algorithms. PR stands for Primal Risk, PD stands for the primal-dual risk and PG stands for the primal gap. NC-$\mu$-SC stands for nonconvex-$\mu$ strongly concave. $\mu$-SC-SC stands for $\mu$ strongly convex-$\mu$ strongly concave. NC-C stands for nonconvex-concave. C-C convex-concave. $L$ is the Lipschitz constant of the function $f$. $\kappa$ stands for the condition number $L/\mu$. The constants in the theorems have been defined in the appropriate sections. Note that there are other results in [Farnia and Ozdaglar, 2021, Lei et al., 2021] for cases where the expectation and $\max$ operator can be interchanged. This case is almost identical to the minimization problem and we thus do not include it in the table.

## A.1 Existing Related Results

From [Farnia and Ozdaglar, 2021], we have the following theorem showing the connection between stability and generalization for minimax problems.

**Theorem 5** ([Farnia and Ozdaglar, 2021]). *Consider an Algorithm A which is $\epsilon$-stable. We have the following two claims:*

1. *If the maximization and the expectation can be swapped when computing $r(w)$, then*

$$E_S E_A[\zeta_{gen}^P(A)] \leq \epsilon.$$

2. *If $f(\cdot, \cdot; z)$ is nonconvex-strongly-concave and $f$ is $\mu$-strongly-concave with respect to $\theta$, then*

$$E_S E_A[\zeta_{gen}^P(A)] \leq L\sqrt{\kappa^2 + 1}\epsilon.$$

**Remark 3.** *In [Lei et al., 2021], the authors proved a generalization bound in a weak sense, i.e., they consider the weak duality gap:*

$$(\max_{\theta \in \Theta} E_S E_A r(w_S^A, \theta) - \min_{w \in W} E_S E_A r(w, \theta_S^A)) - (\max_{\theta \in \Theta} E_S E_A r_S(w_S^A, \theta) - \min_{w \in W} E_S E_A r_S(w, \theta_S^A)).$$

*However, notice that the expectation is inside the min and max operators. It does not deal with the coupling of the maximization and expectation.*

**Remark 4.** *According to Theorem 5, the generalization bound for $\zeta_{gen}^P$ scales with the condition number $\kappa_\theta$, and therefore cannot give useful bounds in the absence of strong concavity (when $\kappa_\theta \to \infty$).*

**Remark 5.** *The generalization bounds for $\zeta_{gen}^P$ of algorithms for problems in terms of stability without strong concavity is still open to the best of our knowledge. As mentioned in [Lei et al., 2021], finding generalization bounds without the strong concavity assumption is an interesting open problem.*

## B When is Primal Risk a Valid Metric for Minimax Learners?

According to the discussions for minimization problems in Section 2, we know that the primal risk is a valid metric to study generalization behavior in these problems, and furthermore, the generalization error bound of the primal risk can be estimated in terms of algorithmic stability. However, Theorem 1 cannot be directly extended to analyze the generalization behavior of minimax learners because we have an additional maximization step before taking expectation.

A natural question emerges: Under what conditions does primal risk serve as a valid metric to study generalization behavior of minimax problems. One sufficient condition is when the maximization step and expectation can be interchanged, i.e., when

$$\max_{\theta \in \Theta} E_{z \sim P_z} f(w, \theta; z) = E_{z \sim P_z}[\max_{\theta \in \Theta} f(w, \theta; z)]$$

for any distribution $P_z$. Letting $f_{\max}(w; z) := \max_{\theta \in \Theta} f(w, \theta; z)$, we further have

$$r(w) = \max_{\theta \in \Theta} E_{z \sim P_z} f(w, \theta; z) = E_{z \sim P_z}[\max_{\theta \in \Theta} f(w, \theta; z)] = E_{z \sim P_z} f_{\max}(w; z).$$

Therefore, the minimax problem in (1) is equivalent to the (stochastic) minimization problem with loss function $f_{\max}(w; z)$. Moreover, letting $P(S)$ be the uniform distribution over the dataset $S = \{z_1, \cdots, z_n\}$, we have

$$r_S(w) = \max_{\theta \in \Theta} E_{z \sim P(S)}[f(w, \theta; z)] E_{z \sim P(S)}[\max_{\theta \in \Theta} f(w, \theta; z)] = \frac{1}{n} \sum_{i=1}^{n} f_{\max}(w; z_i).$$

Therefore, $r_S(w)$ is just the empirical primal risk corresponding to the minimization problem with loss function $f_{\max}(w; z)$. Hence, Theorem 1 can be directly used to minimax problems where the maximization and expectation can be interchanged.

**Theorem 6.** *Suppose that $f(w, \theta; z)$ is $\bar{L}$-Lipschitz continuous with respect to $w$, i.e., $|f(w_1, \theta; z) - f(w_2, \theta; z)| \leq \bar{L}\|w_1 - w_2\|$ for any $w_1, w_2 \in W, \theta \in \Theta$ and $z$. If an Algorithm A is $\epsilon$-stable, we have*

$$E_S E_A[r(w_S^A) - r_S(w_S^A)] \leq \bar{L}\epsilon.$$

*Proof.* From the previous analysis along with Theorem 1, it suffices to show that $f_{\max}(\cdot; z)$ is $\bar{L}$-Lipschitz continuous. In fact, we have

$$f_{\max}(w_1; z) - f_{\max}(w_2; z) = f(w_1, \theta(w_1); z) - f(w_2, \theta(w_2); z)$$
$$\leq f(w_1, \theta(w_1); z) - f(w_2, \theta(w_1); z) \leq \bar{L}\|w_1 - w_2\|,$$

where $\theta(w) \in \arg\max_{\theta \in \Theta} f(w, \theta; z)$, the first inequality is because of the definition of $\theta(w)$ and the second inequality is because of the Lipschitz continuity of $f$ with respect to $w$. Using the same argument, we can prove

$$f_{\max}(w_2; z) - f_{\max}(w_1; z) \leq \bar{L}\|w_1 - w_2\|.$$

Therefore, we prove the $\bar{L}$-Lipschitz continuity of $f_{\max}(\cdot; z)$ and hence finish the proof. $\qquad\square$

By the above discussion, we know that if maximization and expectation can be interchanged, the minimax problem can be reduced to a minimization problem and hence the primal risk is a valid metric for studying the generalization behavior of minimax learners and the generalization error can be estimated using the same method as for minimization problems. In practice, the adversarial-training problems can be such an example of minimax problems.

**Example 4** (Adversarial-training). *We consider the adversarial training problem [Madry et al., 2017]. Suppose we have loss function $g(w; z)$ for a supervised learning problem. Here $z$ denotes the training sample and $w$ denotes the model parameter. Due to the noise in the data or due to an adversarial attack, for any sample $z$, we consider an uncertainty set $B(z, \epsilon_0)$ around it. The goal is to train a model that is robust to the data with possible perturbation in the uncertainty set. Let $\theta_z$ be some adversarial sample from the set $B(z, \epsilon_0)$ and let $\theta$ be an infinite dimensional vector (functional) with the component $\theta_z$ corresponding to the sample $z$. Define the function $\iota_B(v)$ to be the indicator function of the set $B$, i.e., $\iota_B(v) = 0$ if $v \in B$ and $\iota_B(v) = \infty$ otherwise. The goal of adversarial training is to solve the following minimax problem:*

$$\min_{w} \max_{\theta} \ E_{z \sim P_z} f(w, \theta; z), \tag{9}$$

*where $f(w, \theta; z) = g(w; \theta_z) + \iota_{B(z, \epsilon_0)}(\theta_z)$. For any distribution $P_z$ over $z$'s, we have*

$$\max_{\theta} \ E_{z \sim P_z} f(w, \theta; z) = \max_{\theta} \ E_{z \sim P_z}[g(w; \theta_z) + \iota_{B(z, \epsilon_0)}(\theta_z)] = E_{z \sim P_z}[\max_{\theta_z} \ (g(w; \theta_z) + \iota_{B(z, \epsilon_0)}(\theta_z))]$$
$$= E_{z \sim P_z}[\max_{\theta} \ f(w, \theta; z)],$$

*where the second and the third equalities use the fact that $\theta_{z'}$ does not contribute to $f(w, \theta; z)$ if $z \neq z'$. Therefore, the expectation and maximization can be interchanged in adversarial training problems. This implies that the results of Theorem 6 can be applied and therefore primal risk is a valid metric to study the generalization behavior in such problems.*

**Remark 6.** *For Example 4, since the maximization and expectation can be interchanged, the minimax problem is equivalent to a minimization problem. Then we have*

$$E_S[\min_w r_S(w)] = E_S[\min_w \max_\theta E_{z\sim P_z(S)} f(w,\theta;z)] = E_S[\min_w E_{z\sim P_z(S)}[\max_\theta f(w,\theta;z)]]$$
$$= E_S[\min_w E_{z\sim P_z(S)}[f_{\max}(w;z)]] \le E_S[E_{z\sim P_z(S)}[f_{\max}(w;z)]]$$

*for any $w$. Therefore, we have $E_S[\min_w r_S(w)] \le \min_w r(w)$. Consequently, we have $\zeta_{gen}^P \ge \zeta_{gen}^{PG}$, which means that good generalization bounds for the primal risk implies good generalization bounds for the primal gap. Therefore, if the maximization and expectation are interchangeable, primal risk is sufficient to study the generalization behavior because the generalization error of the primal risk is an upper bound of the generalization error of the primal gap in this case.*

Unfortunately, maximization and expectation are not necessarily interchangeable for many minimax problems. If they are not interchangeable, it is unclear how to estimate the generalization error bound of the primal risk. In fact, whether primal risk is still a good metric for studying generalization behavior in such problems remains elusive.

In section 3, we will see how to estimate generalization error bound of primal risk for nonconvex-concave and even nonconvex-nonconcave problems. To the best of our knowledge, this is the first result which provides generalization error bounds for the primal risk without assuming the interchangeability or strong concavity of the inner maximization problems (see e.g., Lei et al. [2021]). Furthermore, we will see that even in some simple minimax problems, the generalization error bound of the primal risk can fail to capture the generalization behavior of minimax learners. We then propose a new metric and use its generalization error to properly characterize the generalization behavior of minimax learners.

## C   Analysis of Example 1

In this section, we analyze the toy example given in Example 1.

**Proposition 2.** *For the risk function and data distribution given in Example 1, we have*

$$E_S[r(w) - r_S(w)] \le 0$$

*for any $w \in W$.*

*Proof.* For a fixed $w$, $r(w) = w^2/2 - w$. On the other hand,

$$r_S(w) = \max_{\theta\in\Theta} r(w,\theta) \tag{10}$$
$$\ge r_S(w,0) \tag{11}$$
$$= r(w). \tag{12}$$

Therefore, we have the desired result. □

Next, we prove that $|\sum_{i=1}^n z_i|$ will stay in the interval $[0.5,\lambda]$ with high probability.

**Lemma 7.** *For large enough $\lambda > 2$, we have*

$$\Pr\left(\left|\sum_{i=1}^n z_i\right| \in [0.5,\lambda]\right) > 0.4, \qquad \Pr\left(\left|\sum_{i=1}^n z_i\right| \in [2,\lambda]\right) > 0.01.$$

*Proof.* Let $y_i \sim N(0,1/\sqrt{n}), i = 1,\cdots,n$ be $n$ i.i.d. variables. Then $\sum_{i=1}^n y_i \sim N(0,1)$. According to the table of Normal distribution, we have $\Pr(|\sum_{i=1}^n y_i| \in [0.5,\lambda]) \ge 0.41$. By the definition of $z_i$, we have

$$\Pr(|\sum_{i=1}^n z_i| \in [0.5,\lambda]) \ge \Pr(|\sum_{i=1}^n y_i| \in [0.5,\lambda], |y_i| < 3\log n/\sqrt{n}) + \Pr(\max_{i\in[n]}(|y_i|) \ge 3\log n/\sqrt{n}).$$

For the first term, we have

$$\Pr(|\sum_{i=1}^n y_i| \in [0.5, \lambda], |y_i| < 3 \log n / \sqrt{n})$$

$$\geq \Pr(|\sum_{i=1}^n y_i| \in [0.5, \lambda]) - \Pr(\max_{i \in [n]}(|y_i|) \geq 3 \log n / \sqrt{n})$$

$$\geq 0.41 - \sum_{i=1}^n \Pr(|y_i| \geq 3 \log n / \sqrt{n})$$

$$\geq 0.41 - n e^{-\gamma 9 \log^2 n} \geq 0.41 - 1/n^{\lambda \gamma - 1}.$$

Taking $\lambda$ sufficiently large yields the desired result, where the first inequality is because of the union bound and the second inequality is due to the tail bound of Normal distribution. Therefore, $\Pr(|\sum_{i=1}^n z_i| \in [0.5, \lambda]) > 0.4$ for sufficiently large $n$. The second statement follows similarly, noting from the table of Normal distribution that $\Pr(|\sum_{i=1}^n y_i| \in [0.5, \lambda]) \geq 0.046$. $\quad\square$

**Proposition 3.** *For sufficiently large $\lambda > 0$, we have*

$$E_S[r(w_S) - \min_{w \in W} r(w)] \geq 0.001.$$

*Proof.* If $|\sum_{i=1}^n z_i| \in [0.5, \lambda]$, we have

$$w_S = \max(0, 1 - (\sum_{i=1}^n z_i)^2 / 2) \leq 0.9.$$

In this case, we have

$$r(w_S) - \min_{w \in W} r(w) \geq 0.005, \tag{13}$$

by direct calculation. Therefore, we have

$$E_S[r(w_S) - \min_{w \in W} r(w)] \tag{14}$$

$$\geq \quad \Pr(|\sum_{i=1}^n z_i| \in [0.5, \lambda]) \cdot 0.05 + \Pr(|\sum_{i=1}^n z_i| \notin [0.5, \lambda]) \cdot 0 \tag{15}$$

$$\geq \quad 0.02, \tag{16}$$

where the first inequality is because of (13) and the fact that $r(w_S) - \min_{w \in W} r(w) \geq 0$ for any $S$. $\quad\square$

**Proposition 4.** *For sufficiently large $\lambda > 0$, we have:*

$$E_S[\min_{w \in W} r_S(w) - \min_{w \in W} r(w)] \geq 0.005$$

*for Example 1.*

*Proof.* If $|\sum_{i=1}^n z_i| \geq \lambda > 2$, we have $w_S = 0$ and hence $r_S(w_S) = 0$. If $|\sum_{i=1}^n z_i| \leq \lambda$, we have

$$r_S(w_S) - r(w^*) \geq r_S(w_S) - r(w_S) = w_S(\sum_{i=1}^n z_i)^2 / 2 \geq 0.$$

Therefore, $\min_{w \in W} r_S(w) \geq \min_{w \in W} r(w)$ for any $S$. By Lemma 7, we can prove that $\Pr(|\sum_{i=1}^n z_i| \in [2, \lambda]) \geq 0.01$ for sufficiently large $\lambda$. Notice that for $|\sum_{i=1}^n z_i| \in [2, \lambda]$, $r_S(w_S) - \min_{w \in W} r(w) = 1/2$. Therefore, we have

$$E_S[\min_{w \in W} r_S(w) - \min_{w \in W} r(w)] \geq \Pr(|\sum_{i=1}^n z_i| \in [2, \lambda]) \cdot 1/2 \geq 0.005.$$

This completes the proof. $\quad\square$

# D Deferred Results and Proofs in Section 3

## D.1 Proof of Lemma 1

In this subsection, we assume that $A$ is an $\epsilon$-stable algorithm. For any $w \in W$, let $\Theta_S(w) = \arg\max_{\theta \in \Theta} r_S(w, \theta)$ and $\Theta(w) = \arg\max_{\theta \in \Theta} r(w, \theta)$ be the solution sets of the problems. Let $\theta(w)$ be any element in $\Theta(w)$. Then

$$
\begin{aligned}
E_A E_S[r(w_S^A) - r_S(w_S^A)] &= E_A E_S[r(w_S^A, \theta(w_S^A)) - r_S(w_S^A, \theta_S(w_S^A))] \\
&\leq E_A E_S[r(w_S^A, \theta(w_S^A)) - r_S(w_S^A, \theta(w_S^A))],
\end{aligned}
$$

where the inequality is because $r_S(w_S^A, \theta_S(w_S^A)) \geq r_S(w_S^A, \theta)$ for any $\theta$. Let $f$ be $\mu$-strongly concave with respect to $\theta$. We denote the condition number by $\kappa_\theta = \ell_{\theta\theta}/\mu$.

In the strongly concave case, $\Theta(w)$ has a unique element $\theta(w)$, which is $\kappa_\theta$-Lipschitz continuous with respect to $w$ (see [Lin et al., 2020]).

Then, defining $\tilde{f}(w, z) = f(w, \theta(w); z)$, the minimax problem reduces to the usual minimization problem on the function $\tilde{f}$. The stability and the Lipschitz continuity of $\theta(w)$ with respect to $w$ yield the generalization bound of $L\sqrt{\kappa^2 + 1}\epsilon$. This is the result shown in Theorem 1 of [Farnia and Ozdaglar, 2021].

However, if the maximization problem is not strongly concave, we lose the Lipschitz continuity and the uniqueness. To overcome this difficulty, we define an approximate maximizer $\bar{\theta}(w)$ to $r(w, \theta)$. Concretely speaking, we define $\bar{\theta}(w)$ to be the point after $s$ steps of gradient ascent for the function $r(w, \cdot)$ with a stepsize $1/\ell_{\theta\theta}$ and being initialized at 0. Then we have the following lemma:

**Lemma 8.** *For any $w \in W$, we have[6]*

1. $\|\bar{\theta}(w) - \bar{\theta}(w')\| \leq s\frac{\ell}{\ell_{\theta\theta}}\|w - w'\|$.

2. $r(w) - r(w, \bar{\theta}(w)) \leq \ell_{\theta\theta}C_p^2/s$.

*Proof.* To prove the first part, let $\theta_0 = \theta_0' = 0$. Define $\theta_t, \theta_t'$ recursively as follows:

$$
\theta_{t+1} = \theta_t + \nabla_\theta r(w, \theta_t)/\ell_{\theta\theta}
$$

and

$$
\theta_{t+1}' = \theta_t' + \nabla_\theta r(w', \theta_t')/\ell_{\theta\theta}.
$$

We prove $\|\theta_t - \theta_t'\| \leq t\frac{\ell}{\ell_{\theta\theta}}\|w - w'\|$ by induction. For $t = 0$, $\theta_0 - \theta_0' = 0$. Assume the induction hypothesis $\|\theta_{t-1} - \theta_{t-1}'\| \leq (t-1)\frac{\ell}{\ell_{\theta\theta}}\|w - w'\|$ holds. We have

$$
\begin{aligned}
\|\theta_t - \theta_t'\| &= \|(\theta_{t-1} + \nabla_\theta r(w, \theta_{t-1})/\ell_{\theta\theta}) - (\theta_{t-1}' + \nabla_\theta r(w, \theta_{t-1}')/\ell_{\theta\theta}) \\
&\quad + (\nabla_\theta r(w, \theta_{t-1}') - \nabla_\theta r(w', \theta_{t-1}'))/\ell_{\theta\theta}\| \\
&\leq \|(\theta_{t-1} + \nabla_\theta r(w, \theta_{t-1})/\ell_{\theta\theta}) - (\theta_{t-1}' + \nabla_\theta r(w, \theta_{t-1}')/\ell_{\theta\theta})\| \\
&\quad + \|(\nabla_\theta r(w, \theta_{t-1}') - \nabla_\theta r(w', \theta_{t-1}'))/\ell_{\theta\theta}\| \\
&\leq \|\theta_{t-1} - \theta_{t-1}'\| + \ell\|w - w'\|/\ell_{\theta\theta} \\
&\leq (t-1)\frac{\ell}{\ell_{\theta\theta}}\|w - w'\| + \frac{\ell}{\ell_{\theta\theta}}\|w - w'\| \\
&= t\frac{\ell}{\ell_{\theta\theta}}\|w - w'\|,
\end{aligned}
$$

where the first inequality follows from the triangle inequality, the second inequality follows from non-expansiveness of gradient ascent for concave functions and the $\ell$-Lipschitz continuity of $\nabla r$, and the third inequality follows from the induction hypothesis.

Therefore, letting $t = s$ completes the proof of the first part. The second part of this lemma is just the convergence result for gradient ascent on smooth concave functions (see e.g., [Nesterov, 2013]). $\square$

---

[6]For point 2, it holds when $s > 0$. For $s = 0$, we have the bound $r(w) - r(w, \bar{\theta}(w)) \leq \ell_{\theta\theta}C_p^2$. We do not separate this degenerate case for ease of presentation.

Consider a virtual algorithm $\bar{A}$: for any $S$, the algorithm returns $w = w_S^A$ and $\theta = \bar{\theta}(w_S^A)$.

**Lemma 9.** *The stability of this virtual algorithm is* $\epsilon\sqrt{\left(s\frac{\ell}{\ell_{\theta\theta}}\right)^2 + 1}$.

*Proof.* It is direct from the first part of Lemma 8. $\qquad\square$

Then we have the generalization bound of $r_S(w, \theta)$:

**Lemma 10.** *We have*

$$E_S E_A[r(w_S^A, \bar{\theta}(w_S^A)) - r_S(w_S^A, \bar{\theta}(w_S^A))] \leq \epsilon L \sqrt{\left(s\frac{\ell}{\ell_{\theta\theta}}\right)^2 + 1}.$$

*Proof.* For any $z$, by Assumption 4, we have

$$\|f(w_S^{\bar{A}}, \theta_S^{\bar{A}}; z) - f(w_{S'}^{\bar{A}}, \theta_{S'}^{\bar{A}}; z)\| \leq \epsilon L \sqrt{\left(s\frac{\ell}{\ell_{\theta\theta}}\right)^2 + 1}.$$

The result follows directly from the standard stability theory in [Hardt et al., 2016]. $\qquad\square$

Now we are ready to derive the generalization error bound of the Primal Risk for an Algorithm $A$ with $\epsilon$-stability. First, we have

$$
\begin{aligned}
E_S E_A[r(w_S^A) - r_S(w_S^A)] &\leq E_S E_A[r(w_S^A) - r_S(w_S^A, \bar{\theta}(w_S^A))] \\
&\leq E_S E_A[(r(w_S^A, \bar{\theta}(w_S^A) + \ell_{\theta\theta}C_p^2/s) - r_S(w_S^A, \bar{\theta}(w_S^A))] \\
&= E_S E_A[r(w_S^A, \bar{\theta}(w_S^A)) - r_S(w_S^A, \bar{\theta}(w_S^A))] + \ell_{\theta\theta}C_p^2/s \\
&\leq \epsilon L \sqrt{\left(s\frac{\ell}{\ell_{\theta\theta}}\right)^2 + 1} + \ell_{\theta\theta}C_p^2/s \\
&\leq \epsilon L s\frac{\ell}{\ell_{\theta\theta}} + \frac{\ell_{\theta\theta}C_p^2}{s} + \epsilon L
\end{aligned}
$$

where the first inequality is because $r_S(w_S^A) = \max_\theta r_S(w_S^A, \theta)$, the second inequality is because of the second part of Lemma 8 and the last inequality is because of Lemma 10. Optimizing over[7] $s$, the generalization error is bounded by $\zeta_{gen}^P(A) \leq \sqrt{4L\ell C_p^2} \cdot \sqrt{\epsilon} + \epsilon L$. This completes the proof. $\qquad\square$

### D.2 Tightness of the bound for Primal Risk

Consider the following risk function:

$$f(w, \theta; z) = \sqrt{n/\epsilon}((w/(n\sqrt{n}\epsilon) - z)\theta - \theta^2/(2n\sqrt{n}\epsilon)),$$

where $w \in W = [-\lambda\epsilon\sqrt{n}\log n, \lambda\epsilon\sqrt{n}\log n]$ and $\theta \in \Theta = \mathbb{R}$. The sample $z$ is drawn from the uniform distribution over $\{-1/\sqrt{n}, 1/\sqrt{n}\}$. Then we have

$$r(w) = \sqrt{n/\epsilon}(w^2/(2\epsilon n\sqrt{n})),$$

and

$$r_S(w) = \sqrt{n/\epsilon}((\epsilon n\sqrt{n})(w/(\epsilon n\sqrt{n}) - \sum_{i=1}^n z_i/n)^2/2).$$

Now we have $\ell = \frac{\sqrt{n/\epsilon}}{(\epsilon n\sqrt{n})}$, $C_p = \lambda\epsilon\sqrt{n}\log n$ and $L = \frac{\sqrt{n/\epsilon}}{\sqrt{n}}$. If we perform one-step of GDMax with stepsize $1/\ell_{r_S}$ where $\ell_{r_S} = \sqrt{n/\epsilon}/(\epsilon n\sqrt{n})$, then we attain $w_S = \arg\min_{w \in W} r_S(w)$. The

---

[7]Here we assume that the optimal $s$ is a real number greater than 0. Constraining $s$ to be an integer and also incorporating 0 does not change the result and we ignore this case here. See also Footnote 6.

stability bound of the GDMax is $\epsilon$. Therefore, the generalization error of the primal risk is estimated as:

$$\zeta_{gen}^P(GDMax) \leq \sqrt{8L\ell C_p^2}\sqrt{\epsilon} = 8\lambda \log n \sqrt{\epsilon}.$$

On the other hand, $\sum_{i=1}^n z_i/n \in [-\lambda \log n/n, \lambda \log n/n]$ holds with probability at least $1 - C/n^\lambda$ by Hoefding inequality. Let $\bar{w}_S = \epsilon\sqrt{n}(\sum_{z_i \in S} z_i)$. Then with probability at least $1 - C/n^\lambda$, $w_S = \bar{w}_S$. Notice that

$$E_S[r(\bar{w}_S) - r_S(\bar{w}_S)] = E_S\left[\sqrt{n} \cdot (\sqrt{\epsilon}n\sqrt{n}) \cdot \left(\sum_{z_i \in S} z_i/n\right)^2\right] = \sqrt{\epsilon}.$$

It is not hard to show that $|r(\bar{w}_S)| \leq n\sqrt{\epsilon}$, $r_S(\bar{w}_S) = 0$, $|r(w_S)| \leq 2n\sqrt{\epsilon}$ and $|r_S(w_S)| \leq 2n\sqrt{\epsilon}$. Then we have

$$E_S[r(\bar{w}_S) - r_S(\bar{w}_S)] - E_S[r(w_S) - r_S(w_S)] \leq 5Cn\sqrt{\epsilon}/n^\lambda.$$

Therefore, $E[r(w_S) - r_S(w_S)] \geq \sqrt{\epsilon}/2$ for sufficiently large $\lambda$ and $n$. Then in this example we have

$$\sqrt{\epsilon}/2 \leq \zeta_{gen}^P(A) \leq 8\lambda \log n \sqrt{\epsilon}.$$

For $\epsilon \leq 1/n^{\tau+1}$, we have

$$\log n \leq \frac{1}{\tau + 1}\log(1/\epsilon).$$

Therefore, the estimate $\zeta_{gen}^P \leq \lambda\sqrt{\epsilon}\log(1/\epsilon)/(\tau + 1)$ is tight up to a $\log(1/\epsilon)$ factor.

### D.3   Proof of Theorem 2

Recall that the empirical primal gap is defined as

$$\Delta_S(w) = r_S(w) - \min_{w \in W} r_S(w)$$

and the population primal gap is given by

$$\Delta(w) = r(w) - \min_{w \in W} r(w).$$

Suppose we are given an $\epsilon$-stable Algorithm $A$. We then want to derive the generalization error

$$\zeta_{gen}^{PG}(A) = E_S E_A[\Delta(w_S^A) - \Delta_S(w_S^A)].$$

Since we already have the generalization error for the primal risk $E_S E_A[r(w_S^A) - r_S(w_S^A)]$ in Theorem 1, we only need to estimate

$$E_S E_A[\min_{w \in W} r_S(w) - \min_{w \in W} r(w)] = E_S[\min_{w \in W} r_S(w) - \min_{w \in W} r(w)]$$

to get a generalization error bound on the primal gap.

**Lemma 11.** *Let $w^* \in \arg\min_{w \in W} r(w)$. Suppose that $f(w^*, \cdot; z)$ is $L_\theta^*$ Lipschitz continuous with respect to $\theta$. Then we have*

$$E_S[\min_{w \in W} r_S(w) - \min_{w \in W} r(w)] \leq 4L_\theta^* C_e/\sqrt{n}.$$

*Proof.* We use similar techniques as in the proof of Lemma 1.

**Step 1.** We define an approximate maximizer $\tilde{\theta}_S$ of the function $r_S(w^*, \cdot)$. $\tilde{\theta}_S$ is attained by performing $s$ steps of gradient ascent to $r_S(w^*, \cdot)$ with stepsize $1/\ell_{\theta\theta}$ and being initialized at 0.

Similar to Lemma 8, we have the following lemma:

**Lemma 12.** *We have the following properties:*

1. $\|\tilde{\theta}_S - \tilde{\theta}_{S'}\| \leq 2sL_\theta^*/(n\ell_{\theta\theta})$.

2. $r_S(w^*) - r_S(w^*, \tilde{\theta}_S) \le \ell_{\theta\theta} C_e^2/s$.

*Proof.* The proof is similar to the proof of Lemma 8. To prove the first part, let $\tilde{\theta}_0 = \tilde{\theta}'_0 = 0$. Define $\tilde{\theta}_t, \tilde{\theta}'_t$ recursively as follows:

$$\tilde{\theta}_{t+1} = \tilde{\theta}_t + \nabla_\theta r_S(w^*, \tilde{\theta}_t)/\ell_{\theta\theta}$$

and

$$\tilde{\theta}'_{t+1} = \tilde{\theta}'_t + \nabla_\theta r_{S'}(w^*, \tilde{\theta}'_t)/\ell_{\theta\theta}.$$

We prove $\|\tilde{\theta}_t - \tilde{\theta}'_t\| \le L_\theta^*/(n\ell_{\theta\theta})$ by induction. For $t = 0$, $\tilde{\theta}_0 - \tilde{\theta}'_0 = 0$. Assume the induction hypothesis $\|\tilde{\theta}_{t-1} - \tilde{\theta}'_{t-1}\| \le (t-1)L_\theta^*/(n\ell_{\theta\theta})$ holds. We have

$$
\begin{aligned}
\|\tilde{\theta}_t - \tilde{\theta}'_t\| &= \|(\tilde{\theta}_{t-1} + \nabla_\theta r_S(w^*, \tilde{\theta}_{t-1})/\ell_{\theta\theta}) - (\tilde{\theta}'_{t-1} + \nabla_\theta r_S(w^*, \tilde{\theta}'_{t-1})/\ell_{\theta\theta}) \\
&\quad + (\nabla_\theta r_S(w^*, \tilde{\theta}'_{t-1}) - \nabla_\theta r_{S'}(w^*, \tilde{\theta}'_{t-1}))/\ell_{\theta\theta}\| \\
&\le \|(\tilde{\theta}_{t-1} + \nabla_\theta r_S(w^*, \tilde{\theta}_{t-1})/\ell_{\theta\theta}) - (\tilde{\theta}'_{t-1} + \nabla_\theta r_S(w^*, \tilde{\theta}'_{t-1})/\ell_{\theta\theta})\| \\
&\quad + \|(\nabla_\theta r_S(w^*, \tilde{\theta}'_{t-1}) - \nabla_\theta r_{S'}(w^*, \tilde{\theta}'_{t-1}))/\ell_{\theta\theta}\| \\
&\le \|\tilde{\theta}_{t-1} - \tilde{\theta}'_{t-1}\| + \ell\|w - w'\|/\ell_{\theta\theta} \\
&\le (t-1)\frac{2L_\theta^*}{n\ell_{\theta\theta}} + \frac{2L_\theta^*}{n\ell_{\theta\theta}} \\
&= t\frac{2L_\theta^*}{n\ell_{\theta\theta}},
\end{aligned}
$$

where the first inequality follows from the triangle inequality, the second inequality follows from non-expansiveness of gradient ascent for concave functions and the $L_\theta^*$-Lipschitz continuity of $f(w^*, \cdot; z)$, and the third inequality follows from the induction hypothesis.

Therefore, letting $t = s$ completes the proof of the first part. The second part of this lemma is just the convergence result for gradient ascent on smooth concave functions (see e.g., [Nesterov, 2013]). □

We then define the virtual algorithm $\tilde{A}$ given by $w_S^{\tilde{A}} = w^*$ and $\theta_S^{\tilde{A}} = \tilde{\theta}_S$. Since the output argument $w$ of $\tilde{A}$ is always $w^*$, the stability of $\tilde{A}$ only depends on $\tilde{\theta}_S$. Then the stability bound of this virtual algorithm is given in the following lemma:

**Lemma 13.** *The stability of Algorithm $\tilde{A}$ is given by $\epsilon_{sta}(\tilde{A}) = 2s(L_\theta^*)^2/(n\ell_{\theta\theta})$.*

Then by the standard stability theory in [Hardt et al., 2016], we have

$$|E_S E_A[r_S(w^*, \tilde{\theta}_S) - r(w^*, \tilde{\theta}_S)]| \le 2s(L_\theta^*)^2/(n\ell_{\theta\theta}). \tag{17}$$

**Step 2.** We have

$$
\begin{aligned}
E_S[\min_{w\in W} r_S(w) - \min_{w\in W} r(w)] &\overset{(i)}{=} E_S[r_S(w_S) - r(w^*, \theta^*)] \\
&\overset{(ii)}{\le} E_S[r_S(w^*) - r(w^*, \theta^*)] \\
&\overset{(iii)}{\le} E_S[r_S(w^*, \tilde{\theta}_S) - r(w^*, \theta^*)] + \ell_{\theta\theta} C_e^2/s \\
&\overset{(iv)}{\le} E_S[r_S(w^*, \tilde{\theta}_S) - r(w^*, \tilde{\theta}_S)] + \ell_{\theta\theta} C_e^2/s,
\end{aligned}
$$

where (i) follows from the definition of $w^*, \theta^*$, (ii) follows since $w_S$ minimizes $r_S(w)$, (iii) follows from Lemma 12, and (iv) follows from the optimality of $\theta^*$ given $w^*$. Then by (17), we have

$$E_S[\min_{w\in W} r_S(w) - \min_{w\in W} r(w)] \le E_S[r_S(w^*, \tilde{\theta}_S) - r(w^*, \tilde{\theta}_S)] + \ell_{\theta\theta} C_e^2/s \tag{18}$$

$$\le 2s(L_\theta^*)^2/(n\ell_{\theta\theta}) + \ell_{\theta\theta} C_e^2/s \tag{19}$$

$$\le 4L_\theta^* C_e/\sqrt{n} \tag{20}$$

which completes the proof. □

The final statement of the theorem follows from Lemma 11 and Lemma 1. □

## D.4 Tightness of the bound for Primal Min Error

We can construct an example with $C_e$, $L_\theta^*$ independent of $n$ and the upper bound for $\zeta_{gen}^{PM}$ is tight up a $\log$ factor.

Consider the following example:

- Let

$$f(w, \theta; z) = \frac{1}{M}(w^2/2 + w(\log^2 n\theta^2/(2K^2) + n\log nz\theta/K + 1)),$$

  where $w \in W = [-1, 1]$, $\theta \in \Theta = [-\lambda K, \lambda K]$ for some arbitrary constants $K > 0$ and $M > 0$. $z$ is drawn from a truncated Gaussian distribution. Concretely speaking, let $y \sim N(0, 1/\sqrt{n})$. Then $z = y$ if $|y| \leq \lambda \log n/\sqrt{n}$ and $z = \lambda \log n/\sqrt{n}$ if $y \geq \lambda \log n/\sqrt{n}$. Notice that $Mf(w, \theta; z) = h(w, \theta'; z)$, where $\theta' = n\log n\theta/K$,

  $$h(w, \theta'; z) = w^2/2 + w((\theta')^2/(2n^2) + z\theta' + 1), \quad \theta' = n\log n\theta/K \in [-\lambda n\log n, \lambda n\log n].$$

  Notice that $h$ is just the risk function in Example 1 in the paper. Therefore, we can estimate the lower bound of the Primal Min Error corresponding to $f$ using the result in Example. The lower bound of Primal Min Error of the problem in Example 1 (corresponding to $h$) is 0.005. Then the Primal-Min Error (corresponding to $f$)

  $$\zeta_{gen}^{PM}(A) = E_S \min_{w \in W} r_S(w) - \min_{w \in W} r(w) \geq 0.005/M.$$

  On the other hand, it is not hard to have $L_\theta^* = \lambda\sqrt{n}\log^2 n/MK$, $C_e = \lambda K$. Therefore, $L_\theta^* C_e = \lambda^2\sqrt{n}\log^2 n/M$. Let $M = \sqrt{n}\log^2 n$. Then $L_\theta^* = \lambda/K$.

  If $K$ does not depend on $n$, $L_\theta^*, C_e$ do not depend on $n$. Therefore, the Primal-Min Error $\zeta_{gen}^{PM}(A)$ satisfies

  $$0.005\lambda^2(L_\theta^* C_e)/(\log^2 n\sqrt{n}) = 0.005(L_\theta^* C_e)/(ML_\theta^* C_e) \leq \zeta_{gen}^{PM}(A) \leq (L_\theta^* C_e)/\sqrt{n},$$

  which is tight up to a factor of $\log^2 n$.

- If we let $M = 1$ and $K = 1$, then $\zeta_{gen}^{PM}(A) \leq \lambda^2\log^2 n$ as discussed in the paper. This upper bound means that we can not attain an arbitrary accuracy $\delta > 0$. The lower bound, i.e., $\zeta_{gen}^{PM}(A) \geq 0.005\lambda^4$, also implies that we can not attain an arbitrary accuracy.

- If we let $M = 1/(\lambda^2\log^2 n)$ and $K = 1$, $L_\theta^* C_e/\sqrt{n} = 1$. This upper bound implies that we can not let $\zeta_{gen}^{PM}$ smaller than arbitrariy required accuracy. Return to the lower bound, i.e., $\zeta_{gen}^{PM}(A) \geq 0.005/(\lambda^2\log^2 n)$. If we want to attain an accuracy $\delta$, the required sample complexity is $2^{\lambda/\sqrt{\delta}}$, which is larger than a polynomial size and hence is still viewed as intractable. In this sense, the upper bound and the lower bound do not make a major difference.

- Combining the two points above, we can conclude that in terms of sample complexity, our bound is tight (up to logarithmic factors).

## D.5 Proof of Lemma 2

We only prove the first part of this lemma and the others can be proved similarly. Let $s = [D_p/\gamma] + 1$, where $[r]$ denotes the largest integer no more than $r$. To prove the first part, let $\theta_0 = \theta_0' = 0$. Define $\theta_t, \theta_t'$ recursively as follows:

$$\theta_{t+1} = \theta_t + c_0\nabla_\theta r(w, \theta_t)/t$$

and

$$\theta_{t+1}' = \theta_t' + c_0\nabla_\theta r(w', \theta_t')/t.$$

We prove $\|\theta_t - \theta'_t\| \le t\frac{\ell}{\ell_{\theta\theta}}\|w - w'\|$ by induction. For $t = 0$, $\theta_0 - \theta'_0 = 0$. Assume the induction hypothesis $\|\theta_{t-1} - \theta'_{t-1}\| \le (t-1)\frac{\ell}{\ell_{\theta\theta}}\|w - w'\|$. We have

$$\|\theta_t - \theta'_t\| = \|(\theta_{t-1} + c_0\nabla_\theta r(w, \theta_{t-1})/t) - (\theta'_{t-1} + c_0\nabla_\theta r(w, \theta'_{t-1})/t) \tag{21}$$

$$+ c_0(\nabla_\theta r(w, \theta'_{t-1}) - \nabla_\theta r(w', \theta'_{t-1}))/t\| \tag{22}$$

$$\le \|(\theta_{t-1} + c_0\nabla_\theta r(w, \theta_{t-1})/t) - (\theta'_{t-1} + c_0\nabla_\theta r(w, \theta'_{t-1})/t)\| \tag{23}$$

$$+ c_0\|(\nabla_\theta r(w, \theta'_{t-1}) - \nabla_\theta r(w', \theta'_{t-1}))/t\| \tag{24}$$

$$\le (1 + c_0\ell_{\theta\theta}/t)\|\theta_{t-1} - \theta'_{t-1}\| + c_0\ell\|w - w'\|/t. \tag{25}$$

Here the first inequality follows from the triangle inequality, the second inequality follows from the $\ell_{\theta\theta}-$Lipschitz continuity of $\nabla_\theta r$ and $\ell$-Lipschitz continuity of $\nabla r$. Therefore, we have

$$\|\theta_t - \theta'_t\| \le (1 + c_0\ell_{\theta\theta}/t)\|\theta_{t-1} - \theta'_{t-1}\| + c_0\ell\|w - w'\|/t.$$

Let $\delta_t = \|\theta_t - \theta'_t\|$. Then by the above recursion, we have

$$\delta_t + \ell/\ell_{\theta\theta}\|w - w'\| \le \prod_{i=1}^{t}(1 + c_0\ell_{\theta\theta}/i)\ell\|w - w'\|/\ell_{\theta\theta}.$$

Using the inequalities $e^a \ge 1 + a$ and $\sum_{i=1}^{t} 1/i \le \log t$, we have

$$\delta_t \le \frac{t\ell}{\ell_{\theta\theta}}\|w - w'\|.$$

Letting $t = s$ yields

$$\|\theta_p^\gamma(w) - \theta_p^\gamma(w')\| \le \frac{s\ell}{\ell_{\theta\theta}}\|w - w'\|.$$

Since $D_p > \gamma$, we have

$$s \le [D_p/\gamma] + 1 \le 2D_p/\gamma.$$

Hence,

$$s\frac{\ell}{\ell_{\theta\theta}} \cdot \gamma \le 2D_p\ell/\ell_{\theta\theta}.$$

Setting $\lambda_p = 2D_p\ell/\ell_{\theta\theta}$ yields the desired result.

### D.6 Proof of Lemma 3

This is similar to the proof of Lemma 1. We first define the virtual algorithm $\bar{A}$ which outputs $(w_S^A, \theta_p^\gamma(w_S^A))$. By Assumption 5, it can be easily seen that $\bar{A}$ is $(1 + \lambda_p/\gamma)\epsilon$-stable. Then by Theorem 1, we have

$$E_S E_A[r(w_S^A, \theta_p^\gamma(w_S^A)) - r_S(w_S^A, \theta_p^\gamma(w_S^A))] \le L(1 + \lambda_p/\gamma)\epsilon.$$

This gives us:

$$E_S E_A[r(w_S^A) - r_S(w_S^A)] \le E_S E_A[r(w_S^A, \theta_p^\gamma(w_S^A)) - r_S(w_S^A, \theta_p^\gamma(w_S^A))] + \gamma$$

$$\le L\epsilon + L\lambda_p\epsilon/\gamma + \gamma.$$

Taking $\gamma = \sqrt{L\lambda_p}\sqrt{\epsilon}$, we have

$$\zeta_{gen}^p(A) \le L\epsilon + \sqrt{L\lambda_p}\sqrt{\epsilon}.$$

## E   Proofs in Section 4

### E.1   Proof of Theorem 4

First, we have

$$E_S E_{A_0}[r(w_S^{A_0,T}) - \min_{w \in W} r(w)]$$

$$= E_S E_{A_0}[r_S(w_S^{A_0,T}) - \min_{w \in W} r_S(w)] + E_S E_{A_0}[r(w_S^{A_0,T}) - r_S(w_S^{A_0,T})]$$

$$+ E_S E_{A_0}[\min_{w \in W} r_S(w) - \min_{w \in W} r(w)]. \tag{26}$$

---

**Algorithm 1** GDA

---

**Input:** initial iterate $(w_S^0, \theta_S^0) = (0, 0)$, stepsizes $\alpha_t, \beta_t$, projection operators $P_W$ and $P_\Theta$;
1: **for** $t = 0, \dots, T - 1$ **do**
2:    $w_S^{t+1} = P_W\left(w_S^t - \alpha_t \nabla_w r_S(w, \theta)\right)$
3:    $\theta_S^{t+1} = P_\Theta\left(\theta_S^t + \beta_t \nabla_\theta r_S(w, \theta)\right)$
4: **end for**

---

---

**Algorithm 2** GDMax

---

**Input:** initial iterate $(w_S^0, \theta_S^0) = (0, 0)$, stepsizes $\alpha_t$, projection operators $P_W$ and $P_\Theta$;
1: **for** $t = 0, \dots, T - 1$ **do**
2:    $w_S^{t+1} = P_W\left(w_S^t - \alpha_t \nabla_w r_S(w, \theta)\right)$
3:    $\theta_S^{t+1} = \underset{\theta \in \Theta}{\arg\max}\, r_S(w_S^{t+1}, \theta)$
4: **end for**

---

Furthermore, by Assumption 7 and Theorem 2, we have

$$E_S E_{A_0}[r(w_S^{A_0, T}) - \min_{w \in W} r(w)] \le (\phi_{A_0}(M_w) + \phi_{A_0}(C_e(\Theta_0)))/\psi_{A_0}(T) + \zeta_{gen}^P(A_0) + L_\theta^* C_e(\Theta_0)/\sqrt{n}.$$

Next, notice that the output of $A_0$ is equal to the output of $A$ with probability at least $1 - \delta$ and $\|r(w)\| \le 1$. Therefore, we have

$$|E_S E_A[r(w_S^{A, T})] - E_S E_{A_0}[r(w_S^{A_0, T})]| \le \delta,$$

which gives the desired result. $\qquad\square$

### E.2 Proof of Lemma 4

Define $\delta_t = \|(w_S^t, \theta_S^t) - (w_{S'}^t, \theta_{S'}^t)\|$. We have
$$\delta_{t+1} \le (1 + c_0 \ell/t)\delta_t + 2c_0 L_{\Theta_\theta}^{GDA}/nt.$$

Therefore,

$$\delta_{t+1} + \frac{2L_{\Theta_\theta}^{GDA}}{\ell n} \le (1 + c_0 \ell/t)\left(\delta_t + \frac{2L_{\Theta_\theta}^{GDA}}{\ell n}\right) \le \frac{2L_{\Theta_\theta}^{GDA}}{\ell n} T^{c_0 \ell}, \qquad (27)$$

which completes the proof. $\qquad\square$

### E.3 Proof of Lemma 5

For a fixed dataset $S$, let $g_t = \nabla r_S(w^t, \theta^t)$ and $d_t = \|(w^0, \theta^0) - (w^t, \theta^t)\|$. Then we have $g_t \le L_0 + d_t \ell$ and $d_{t+1} \le d_t + c_0 g_t/t$. Substituting the first inequality into the second one, we have
$$d_{t+1} \le d_t + c_0 d_t/t + L_0 c_0/t,$$

which gives us
$$d_{t+1} + L/\ell \le (1 + c_0 \ell/t)(d_t + L_0/\ell).$$
Multiplying this inequality from 0 to $T - 1$ yields
$$d_T \le T^{c_0 \ell} L_0/\ell,$$

which completes the proof. $\qquad\square$

### E.4 Proof of Lemma 6

Let $u = [1, 1, \cdots, 1, 0, \cdots, 0]^T \in \mathbb{R}^{2n}$. Then $\theta_S(w)$ satisfies $Q_S^T \theta_S(w) = u - b_0 e$, where $e = [1, 1, \cdots, 1]^T \in \mathbb{R}^{2n}$. It can be easily seen that $\|u - b_0 e\| \ge \sqrt{n}/2$.

We can also show that $\sigma_{\max}(Q_S) \le 2\sigma_{\max} \cdot \sigma_{\max}(P)$, where $P \in \mathbb{R}^{2n \times m}$ is full row-rank and independent rows. Moreover, every row of $P$ has covariance matrix $I_m/\sqrt{m}$. Then by random matrix theory (see [Vershynin, 2010]), we have $\sigma_{\max}(P) \le \mathcal{O}(\sqrt{m}/\sqrt{m} - C\sqrt{n}/\sqrt{m} + \log(1/\delta)/\sqrt{m}) = \mathcal{O}(1)$ with probability $1 - C\delta$. Therefore, we have $\theta_S(w) \ge \Omega(\sqrt{n})$. $\qquad\square$

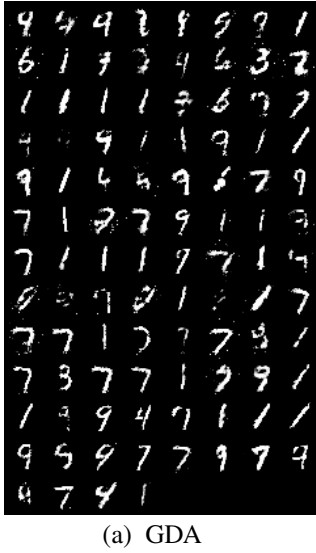
(a) GDA

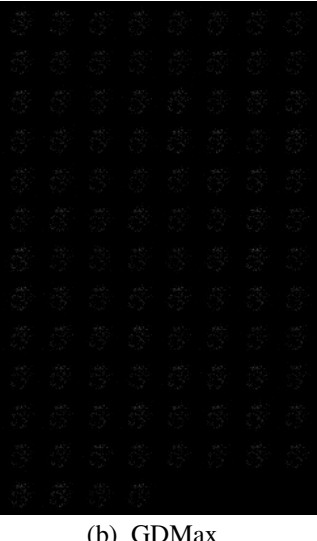
(b) GDMax

Figure 1: Comparison of the results on MNIST generated by GDA and GDMax.

### E.5 Experiments on GAN-training

In this section, we provide some numerical results to corroborate our theoretical findings.

#### E.5.1 Setup

We train a GAN on MNIST data using two algorithms – GDA and GDMax. Since the stability is improved by using adaptive methods like Adam, we use Adam-descent-ascent (ADA) and Adam-descent-max (ADMax) instead. ADA simultaneously trains the generator and the discriminator, while ADMax trains the optimal discriminator for each generator step. We simulate this by taking 10 steps of ascent for every descent step. Figure 1 plots the images generated by GANs trained using these two algorithms. Finally, in Figure 2, we plot the norms of the discriminator trained by these two algorithms.

#### E.5.2 Results

Figure 1 plots the images generated by GANs trained using GDA and GDMax (using Adam instead of the simple gradient step). As predicted by the theory in Section E, we can see that GDA produces better images than the corresponding GAN trained using GDMax. Furthermore, the claim that $C_e >> C_p$ can be seen from Figure 2 where we see that the norm of the discriminator trained using GDMax is much larger than the norm of the discriminator trained using GDA. This follows from the results in Section 4.2. GDMax trains the discriminator to exactly distinguish between the empirical data generated by the true and fake distributions. Therefore, when they are nearly the same, their empirical distributions would be close as well. This would imply that the discriminator would need to have a very large slope (Lipschitz constant) to exactly distinguish between the two empirical datasets, and this in turn leads to a large discriminator norm (which captures the Lipschitz constant of the discriminator).

## F Generalization Error for Primal-Dual Risk

If the saddle-point exists, the *primal-dual risk* is often a good measure of generalization:

**Definition 7.** *[Primal-dual risk] The population and empirical primal-dual (PD) risks are defined as:*

$$\Delta^{PD}(w, \theta) = \max_{\theta' \in \Theta} r(w, \theta') - \min_{w' \in W} r(w', \theta),$$

*and*

$$\Delta_S^{PD}(w, \theta) = \max_{\theta' \in \Theta} r_S(w, \theta') - \min_{w' \in W} r_S(w', \theta).$$

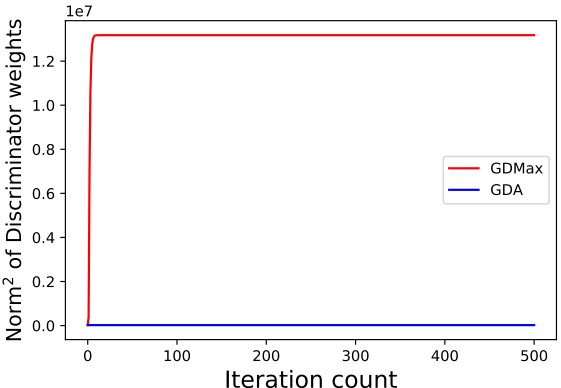

Figure 2: Comparison of the norm squares of discriminator weights.

A point $(w, \theta)$ is called a saddle-point of $r_S$ (or $r$) if $\Delta_S^{PD}(w, \theta) = 0$ (or $\Delta^{PD}(w, \theta) = 0$). Furthermore, if a saddle-point $(w_S, \theta_S)$ exists for $r_S(\cdot, \cdot)$, we have $w_S = \min_{w \in W} r_S(w)$. Moreover, if $w_S \in \arg\min_{w \in W} r_S(w)$ and $\theta_S \in \arg\max_{\theta \in \Theta} r_S(w_S, \theta)$, then $(w_S, \theta_S)$ is a saddle point of $r_S(\cdot, \cdot)$.

Notice that if we can find an approximate saddle point $(w_S, \theta_S)$ of $r_S(w, \theta)$, i.e., $\Delta_S^{PD}(w_S, \theta_S) < \epsilon$ and guarantee that $\Delta^{PD}(w_S, \theta_S) - \Delta_S^{PD}(w_S, \theta_S)$ is small, we can guarantee that $\Delta(w_S, \theta_S)$ is small and therefore $(w_S, \theta_S)$ is an approximate saddle point of $r(\cdot, \cdot)$. Hence if the saddle point exists for $r_S(\cdot, \cdot)$, the generalization error of the primal-dual risk can be a good measure for the generalization of the solution to the empirical problem. We define the expected generalization error for the primal-dual risk as follows:

**Definition 8.** *The generalization error for the primal-dual risk is defined as*

$$\zeta_{gen}^{PD}(A) = E_S E_A[\Delta^{PD}(w_S^A, \theta_S^A) - \Delta_S^{PD}(w_S^A, \theta_S^A)].$$

### F.1 The generalization of the primal-dual risk for convex-concave problems

Similar to Definition 6, we define the $W$-capacity as follows:

**Definition 9** (W-Capacity). *Let*

$$W^*(\theta) = \min_{w \in W} r(w, \theta), \quad and \quad W_S(\theta) = \min_{w \in W} r_S(w, \theta).$$

*The $W$-capacities $C_e^w$ and $C_p^w$ are defined as*

$$C_p^w = \max_{\theta} \text{dist}(0, W^*(\theta)$$
$$C_e^w = \max_{S, \theta} \text{dist}(0, W_S(\theta)). \tag{28}$$

Next, we also define the following:

**Definition 10.** *Let $f^-(\theta, w; z) = -f(w, \theta; z)$. We first have*

$$r^-(\theta, w) = E_{z \sim P_z}[f^-(\theta, w; z)], \qquad r_S^-(\theta, w) = \frac{1}{n} \sum_{i=1}^{n} f^-(\theta, w; z_i). \tag{29}$$

*Furthermore, we define:*

$$r^-(\theta) = \max_{w \in W} r^-(\theta, w) = -(\min_{w \in W} r(w, \theta))$$
$$r_S^-(\theta) = \max_{w \in W} r_S^-(\theta, w) = -(\min_{w \in W} r_S(w, \theta)). \tag{30}$$

Now, we have the following bound for the generalization error of the primal-dual risk, $\zeta_{gen}^{PD}(A)$ for an $\epsilon$-stable Algorithm $A$:

**Theorem 7.** *Suppose that Algorithm $A$ is $\epsilon$-stable. The generalization error $\zeta_{gen}^{PD}(A)$ for convex-concave problem, i.e., when $f(\cdot, \cdot; z)$ is convex-concave for all $z$, is bounded by:*

$$\zeta_{gen}^{PD}(A) \leq \left( \sqrt{4L\ell C_p^2} + \sqrt{4L\ell (C_p^w)^2} \right) \sqrt{\epsilon} + 2\epsilon L.$$

*Proof.* Notice that

$$
\begin{aligned}
\zeta_{gen}^{PD}(A) &= E_S E_A[\Delta^{PD}(w_S^A, \theta_S^A) - \Delta_S^{PD}(w_S^A, \theta_S^A)] \qquad (31) \\
&= E_S E_A[r(w_S^A) - r_S(w_S^A)] + E_S E_A[r^-(\theta_S^A) - r_S^-(\theta_S^A)]. \qquad (32)
\end{aligned}
$$

The two terms can be bounded by Lemma 1 respectively. By Lemma 1, we have

$$E_S E_A[r(w_S^A) - r_S(w_S^A)] \leq \sqrt{4L\ell C_p^2}\sqrt{\epsilon} + \epsilon L$$

and

$$E_S E_A[r^-(\theta_S^A) - r_S^-(\theta_S^A)] \leq \sqrt{4L\ell(C_p^w)^2}\sqrt{\epsilon} + \epsilon L.$$

Combining these two inequalities yields the desired result. $\qquad \square$

## F.2   $\zeta_{gen}^{PD}(T)$ for the proximal point algorithm

In this section, we study the generalization behavior of the proximal point algorithm (PPA) ((See Equation (3) in [Farnia and Ozdaglar, 2021])). By [Farnia and Ozdaglar, 2021], the stability of $T$ steps of PPA can be bounded as follows:

**Lemma 14** ([Farnia and Ozdaglar, 2021]). *The stability of $T$ steps of PPA can be bounded by $\epsilon \leq \mathcal{O}(T/n)$.*

Therefore, substituting the result of Lemma 14 in Theorem 7, we have the following bound for $\zeta_{gen}$ for $T$ steps of PPA:

**Theorem 8.** *After $T$ steps of PPA, the generalization error of the primal-dual risk can be bounded by:*

$$\zeta_{gen}^{PD}(T) \leq \mathcal{O}\left( \sqrt{T/n} + T/n \right).$$

## F.3   The population primal-dual risk of PPA

Finally, we give the population primal-dual risk after $T$ steps of PPA. By [Mokhtari et al., 2020b], we have the following convergence result of PPA.

**Lemma 15** ([Mokhtari et al., 2020b]). *Let $(w_S^t, \theta_S^t)$ be the iterates obtained after $t$ iterations of proximal point algorithm on the function $r_S(\cdot, \cdot)$ and $\bar{w}_S^t = \frac{1}{t}\sum_{i=1}^t w_S^i, \bar{\theta}_S^t = \frac{1}{t}\sum_{i=1}^t \theta_S^i$ be the averaged iterates. Then we have*

$$\Delta_S^{PD}(\bar{w}_S^T, \bar{\theta}_S^T) \leq \ell(C_e^2 + (C_e^w)^2)/T.$$

Combining Lemma 15 and Theorem 8, we have the following result:

**Theorem 9.** *Let $(w_S^t, \theta_S^t)$ be the iterates obtained after $t$ iterations of proximal point algorithm on the function $r_S(\cdot, \cdot)$ and $\bar{w}_S^t = \frac{1}{t}\sum_{i=1}^t w_S^i, \bar{\theta}_S^t = \frac{1}{t}\sum_{i=1}^t \theta_S^i$ be the averaged iterates. Then, the expected population primal-dual risk at the point $(\bar{w}_S^t, \bar{\theta}_S^t)$ can be bounded by:*

$$E_S[\Delta^{PD}(w_S^t, \theta_S^t)] \leq \mathcal{O}\left( 1/T + \sqrt{T/n} + T/n \right).$$