# OpenReview forum: "What is a Good Metric to Study Generalization of Minimax Learners?"
_NeurIPS.cc/2022/Conference — NeurIPS 2022 Accept_

### Official Review · Reviewer_hkx3 · 2022-06-28

**Rating:** 6
**Confidence:** 4
**Soundness:** 3 good
**Presentation:** 3 good
**Contribution:** 2 fair

**Summary:**

This paper introduces an example to show that the primal risk metric has inadequacies in studying the generalization for minimax problems.

This paper then proposes a new metric – the primal gap to avoid the issue in the example.

This paper then leverages this new metric to compare the generalization behavior of GDA and GDMax.

Finally, this paper establishes generalization bounds for primal risk and primal-dual risk without assuming strong concavity.

**Questions:**

For the minimax problem, good generalization behavior is defined on the primal function. My first question is why good generalization behavior should be defined on the primal function? Is this necessary and is this a unique pattern to measure the good generalization behavior for the minimax problem.

The authors provide the primal gap as a new metric to replace the primal risk. Although there have some explanations below Definition 4, I still do not understand why this metric is a good one. Can the authors provide a clear explanation?

**Limitations:**

The authors have adequately addressed the limitations and potential negative societal impact of their work.



**Strengths And Weaknesses:**

Strengths:

The introduction is very clear.

The generalization without assuming strong concavity has good contributions.

Weaknesses:

Does Theorem 1 (d) of the paper [1] study a more general stability measure of the primal gap? Can the authors provide a discussion?

Missing reference:

[1] Li, S. and Liu, Y. High probability generalization bounds with fast rates for minimax problems. In ICLR, 2022.

---

> ### Author Response · Authors · 2022-08-02
> **Response to Reviewer hkx3**
>
> We thank the reviewer for the detailed comments. Please find the response below:
>
> **Does Theorem 1 (d) of the paper [1] study a more general stability measure of the primal gap? Can the authors provide a discussion?**
>
> Thank you very much for bringing up this nice reference. We were not aware of the paper while submitting our work. We will add a detailed discussion about the reference in the final version. We now summarize the difference as follows:
>
>
> 1) Theorem 1(d) of [Li and Liu, 2022] provides a bound for the primal-dual risk for strongly-convex-strongly concave functions. We analyze this metric in Section E of the appendix. We in fact provide a bound for primal-dual risk for the convex-concave case (in Theorem 7) which is the first bound on primal-dual risk without strong convexity nor interchangeability of max and expectation, to the best of our knowledge.
>
> 2) The work [Li and Liu, 2022] discussed several different stability measures. However, discussing different stability measures is not our main focus. We only use the framework of  stability to find an appropriate metric of  generalization for minimax learners. Using our framework to analyze more algorithms is an interesting future direction, which would require more discussion on different stability measures.
>
> 3) Furthermore, as we describe in the response to Reviewer \textit{Z6Bm}, primal-dual risk is not good metric in general to measure generalization in nonconvex settings, since a saddle-point, i.e., a point with the primal-dual risk exactly equal to $0$, may not exist.
>
> 4) In [Li and Liu, 2022], the authors improved the higher probability bound for generalization error while we mainly focus on expected generalization bounds in. It is an interesting and important future direction to extend our results to high probability ones. Combining the techniques in our paper with those in [Li and Liu, 2022] should give sharper results in this new metric and relaxed convexity setting.
>
>
> **My first question is why good generalization behavior should be defined on the primal function? Is this necessary and is this a unique pattern to measure the good generalization behavior for the minimax problem:**
>
> 1) The main motivation for this comes from the  minimax problems in  machine learning such as GAN and adversarial training.   In most of these settings such as GAN, we are only interested finding a good primal variable and the dual variable can be viewed as an auxiliary  variable. So the goal of learning in these problems is to **minimize the population primal risk**  function. In this sense, the good generalization for primal is enough. We do not need to study the generalization of the **dual risk** (defined accordingly).
>
> 2) Moreover, if the saddle-point does not exist for minimax problems (this usually happens in nonconvex cases, which are ubiquitous in machine learning problems), the  popular measure -- the primal-dual risk -- fails to capture the optimality of a solution and hence we can not use it if the saddle-point does not exist (see response to Reviewer \textit{Z6Bm}), which would happen in GAN training, e.g., see [1].
>
> 3) It is definitely not the **unique pattern** to measure the good generalization behavior for minimax learners. For the settings where the  saddle point exists (and the primal and dual variables play symmetric  roles), the primal-dual risk can be a good metric. But interestingly, note that in this case, there is a close connection between **primal-dual risk** and **primal gap** we proposed.
>
>     Considering the equivalent  minimax problem (due to the existence of saddle points) $\min_{\theta\in \Theta}\max_{w\in W}(-r(w,\theta))$, the original  dual variable $\theta$ becomes primal variable of this  problem. We define the primal gap for the variable $\theta$ in this equivalent problem to be the dual gap of the original problem, which is denoted as $\delta^D(\theta)$. Concretely speaking, $$\Delta(\theta)=\max_{\theta\in \Theta}\min_{w\in W}r(w,\theta)-\min_{w\in W}r(w,\theta).$$
>
>
>     In this case, it is not hard to see that the primal-dual risk is just the sum of the primal gap and dual gap: $$\Delta^{PD}(w,\theta)=\Delta(w)+\Delta^D(\theta).$$ So if the saddle-point exists, the primal dual gap is just the sum of the primal gaps corresponding to the primal and dual variables. This means that the derivation for the generalization bound for primal gap (and symmetrically for dual gap) can be used to derive that for primal-dual risk in this case.
>
> 5) Even in terms of primal-dual risk, we have improved the generalization bound compared to the literature, since we developed it for the convex-concave case which has been left as open in the literature.

---

> > ### Author Response · Authors · 2022-08-02
> > **Response to Reviewer hkx3 (cont')**
> >
> > **I still do not understand why this metric is a good one. Can the authors provide a clear explanation?**
> >
> > We summarize why our metric is good as follows.
> >
> > 1) A good generalization bound $\zeta(A)$ should satisfy: if $E_SE_A[r_S(w_S^A)-\min_{w\in W}r_S(w)]$ is small, then $E_SE_A[r(w_S^A)-\min_{w\in W}r(w)]$ is small if and only if $\zeta(A)$ is small.
> >
> > 2) From Example 1, we see that the generalization bound of the primal risk for algorithm $A$, i.e., $\zeta_{gen}^P(A)$, is not a good generalization bound. This is because in Example 1, $r_S(w_S^A)=\min_{w\in W}r_S(w)$ and $\zeta_{gen}^P(A)$ is small, but $E_SE_A[r(w_S^A)]-\min_{w\in W}r(w)$ is highly suboptimal (constant error). Unfortunately, in the literature, no paper notices that primal risk  and primal-dual risk are not enough to study all kinds of minimax problems (especially those in machine learning applications that contains nonconvexity).
> >
> > 3) We have the decomposition
> >
> > $$
> > E_SE_A[r(w_S^A)-\min_{w\in W}r(w)]\\
> > =E_SE_A[r_S(w_S^A)-\min_{w\in W}r_S(w)]+E_SE_A[(r(w_S^A)-\min_{w\in W}r(w))-(r_S(w_S^A)-\min_{w\in W}r_S(w))]\\
> > =E_SE_A[r_S(w_S^A)-\min_{w\in W}r_S(w)]+\zeta_{gen}^{PG}(A).
> > $$
> >
> > If $E_SE_A[r_S(w_S^A)-\min_{w\in W}r_S(w)]$ is small and $\zeta_{gen}^{PG}(A)$ is small (or large), then $E_SE_A[r(w_S^A)-\min_{w\in W}r(w)]$ is small (or large).
> >
> > 4) For Example 1, $\zeta_{gen}^P(A)\le 0$ for algorithm $A$ but $\zeta_{gen}^{PG}(A)\ge 0$. We know that $A$ has constant suboptimal gap for the population risk $r(w_S^A)$. Therefore, $\zeta_{gen}^{PG}(A)$ can imply the bad behavior of algorithm but $\zeta_{gen}^P(A)$ can not.
> >
> >
> > We hope our responses have addressed all your concerns. We have also added these explanations in our revision.
> >
> >
> >
> > References:
> >
> > [1] Farnia, F., \& Ozdaglar, A. (2020, November). Do GANs always have Nash equilibria?. In International Conference on Machine Learning (pp. 3029-3039) PMLR.

---

> > > ### Comment · Reviewer_hkx3 · 2022-08-10
> > > **Sorry for the delayed response.**
> > >
> > >
> > > Thank the authors for the detailed response. The response has addressed my concerns well. I increase my score to 6 considering this paper has made solid contributions to nonconvex minimax problems.
> > >
> > > minor issue:
> > >
> > > Line 193: missing a pair of parentheses

---

### Official Review · Reviewer_VSNm · 2022-07-01

**Rating:** 6
**Confidence:** 3
**Soundness:** 3 good
**Presentation:** 4 excellent
**Contribution:** 3 good

**Summary:**

The paper proposes a new metric, the primal gap, to study the generalization of minimax learners. The paper gives an overview of shortcomings of previous metrics for minimization problems (the primal risk and primal-dual risk) when they are applied to minimax problems. Then, it is shown that the primal gap metric works well specifically for minimax problems. Generalization bounds are derived, and as special cases applied to primal risk and primal-dual risk for the nonconvex-concave setting, showing new bounds there. Finally, comparisons between gradient descent-ascent and gradient descent-max are given.

**Questions:**

* Is the dependence of $\sqrt{\epsilon}$ optimal? It appears to be inconsistent for the (albeit different) setting under strong convexity.
* How do the parameters $C_p$ and $C_e$ affect dependence on $\epsilon$ and $n$? The example given in 3.4 shows that factors relating to $n$ may appear.
* Following the above question, from the examples given in Section 4, $C_p$ and $C_e$ play a key role in comparing GDA and GDMax. Will a similar reasoning throughout the paper, when applied to the other settings given in Table 1, replace dependence of kappa and introduce terms related to $C_p$ and $C_e$? Are there further comparisons between the main results and the previous work?

**Limitations:**

Limitations are not adequately addressed within the paper; see questions above.

**Strengths And Weaknesses:**

Overall, the results are interesting and the paper is nicely written.

**Originality.**
The paper proposes a new metric, the primal gap, for the generalization of minimax learners. The primal gap appears to be a slightly adjusted version of primal risk and primal-dual risk for the minimax setting. The paper discusses several generalization bounds under the primal gap --- see below sections for questions regarding comparisons with other work.

**Quality and clarity.**
The paper is nicely written and results are stated clearly. There are some minor typos that are listed here for further revision:
- Table 1 contains a possible error: "in the in the theorems"
- Line 270: "generalizationerror"

**Significance.**
The study of minimax learning is important and the paper brings interesting results into this area. However, there are several issues. After the proposal of the primal gap, the main result is Lemma 1 and Theorem 2, but the example given in Section 3.4 is not very convincing. While the gap between the lower and upper bounds is only a log-factor, which is often ignored, in this case a difference between a constant factor and a log-factor is quite large. Further, it is unclear whether the dependence of $\sqrt{\epsilon}$ is optimal or not --- particularly in the regime with small $\epsilon$, it appears there is a large gap between the results here and previous results although they are given under stronger convexity constraints. The dependence on $C_p$ and $C_e$ is also not clearly addressed, as it appears factors of $n$ may be introduced, as evidenced by the example in Section 3.4.

---

> ### Author Response · Authors · 2022-08-02
> **Response to Reviewer VSNm**
>
> We thank the reviewer for the  detailed comments. We have corrected the typos you pointed out. Please find the detailed response below:
>
> **The dependence on $C_p$ and $C_e$ is also not clearly addressed, as it appears factors of  n may be introduced, as evidenced by the example in Section 3.4.**
>  We note that, in general,
> $C_e$ and $C_p$ do not necessarily depend on $n$. The Example 1 is just a special case, where we let $C_e=\lambda n
> \log n$ to explain the role of the Primal-Min Error.
>
>
> **Example given in Section 3.4 is not very convincing.**
> We fully understand your concerns.  However, we still think it is convincing, and give more explanations as follows:
>
> 1) We can construct an example with $C_e$,$L_{\theta}^*$ independent of $n$ and the upper bound for $\zeta_{gen}^{PM}$ is tight up a $\log$ factor.
> Consider the following example:
> Let $f(w,\theta;z)=\frac{1}{M}(w^2/2+w(\log^2n\theta^2/(2K^2)+ n\log nz\theta/K+1))$, where $w\in W=[-1,1]$, $\theta\in \Theta=[-\lambda K,\lambda K]$ for some arbitrary constants  $K>0$ and $M>0$. $z$ is drawn from a  truncated Gaussian distribution. Concretely speaking, let $y\sim N(0,1/\sqrt{n})$. Then $z=y$ if $|y|\le \lambda \log n/\sqrt{n}$ and $z=\lambda \log n/\sqrt{n}$ if $y\ge \lambda \log n/\sqrt{n}$.
> Notice that $Mf(w,\theta;z)=h(w,\theta';z)$, where $\theta'=n\log n \theta/K$, $h(w,\theta';z)=w^2/2+w((\theta')^2/(2n^2)+z\theta'+1)$, $\theta'=n\log n \theta/K\in [-\lambda n\log n,\lambda n\log n]$.
> Notice that  $h$ is just the risk function in Example 1 in the paper. Therefore, we can estimate the lower bound of the Primal Min Error corresponding to $f$  using the result in Example.
> The lower bound of Primal Min Error of the problem in Example 1 (corresponding to $h$) is $0.005$. Then the Primal-Min Error  (corresponding to $f$) $\zeta_{gen}^{PM}(A)=E_S\min_{w\in W}r_S(w)-\min_{w\in W}r(w)\ge 0.005/M$.
> On the other hand, it is not hard to have $L_{\theta}^*=\lambda\sqrt{n}\log^2n/MK$, $C_e=\lambda K$. Therefore,
>  $L_{\theta}^*C_e=\lambda^2\sqrt{n}\log^2n/M$.
>
>     Let $M=\sqrt{n}\log^2n $. Then $L_{\theta}^*=\lambda/K$.
>
>     If $K$ does not depend on $n$, $L_{\theta}^*, C_e$ do not depend on $n$. Therefore, the Primal-Min Error $\zeta_{gen}^{PM}(A)$ satisfies $0.005\lambda^2(L_\theta^*(C_e))/(\log^2n \sqrt{n})=0.005(L_\theta^*(C_e))/(ML_\theta^*(C_e))\le \zeta_{gen}^{PM}(A)\le (L_\theta^*(C_e))/\sqrt{n}$, which is tight up to a $\log^2 n$.
>
>
> 2) If we let $M=1$ and $K=1$, then $\zeta_{gen}^{PM}(A)\le \lambda^2\log^2n$ as discussed in the paper. This upper bound means that we can not attain an arbitrary accuracy $\delta>0$.  The lower bound, i.e., $\zeta_{gen}^{PM}(A)\ge 0.005\lambda^4$, also implies that we can not attain  an arbitrary accuracy.
>
> 3) If we let $M=1/(\lambda^2\log^2n)$ and $K=1$, $L_{\theta}^*C_e/\sqrt{n}=1$. This upper bound implies that we can not let $\zeta_{gen}^{PM}$ smaller than arbitrariy required accuracy. Return to the lower bound, i.e., $\zeta_{gen}^{PM}(A)\ge 0.005/(\lambda^2\log^2n)$. If we want to attain an accuracy $\delta$, the required sample complexity is $2^{\lambda/\sqrt{\delta}}$, which is larger than a polynomial size and hence is still viewed as intractable. In this sense, the upper bound and the lower bound do not make a major difference.
>
> 4) Combining the two points above, we can conclude  that in terms of sample complexity, our bound is tight (up to logarithmic factors).
>
>
> We have included this in Appendix Section C.4.

---

> > ### Author Response · Authors · 2022-08-02
> > **Response to Reviewer VSNm (cont.)**
> >
> > **Is $\sqrt{\epsilon}$ optimal?:** Thanks for the careful reading and enlightening comment. Inspired by your comment, we show in the following example that this $\sqrt{\epsilon}$ dependence **is tight**.
> >
> > Consider the following risk function: $f(w,\theta;z)=\sqrt{n/\epsilon}((w/(n\sqrt{n}\epsilon)-z)\theta-\theta^2/(2n\sqrt{n}\epsilon))$,
> > where $w\in W=[-\lambda \epsilon\sqrt{n}\log n,\lambda \epsilon\sqrt{n}\log n]$ and $\theta\in \Theta=\mathbb{R}$.
> > The sample $z$ is drawn from the uniform distribution over {$-1/\sqrt{n},1/\sqrt{n}$}.
> > Then we have $r(w)=\sqrt{n/\epsilon}(w^2/(2\epsilon n\sqrt{n}))$ and $r_S(w)=\sqrt{n/\epsilon}((\epsilon n \sqrt{n})(w/(\epsilon n\sqrt{n})-\sum_{i=1}^nz_i/n)^2/2)$.
> > Now we have $\ell=\frac{\sqrt{n/\epsilon}}{(\epsilon n\sqrt{n})}$, $C_p=\lambda \epsilon\sqrt{n}\log n$ and $L=\frac{\sqrt{n/\epsilon}}{\sqrt{n}}$. If we perform one-step of GDMax with stepsize $1/\ell_{r_S}$ where $\ell_{r_S}=\sqrt{n/\epsilon}/(\epsilon n\sqrt{n})$, then we attain $w_S=\arg\min_{w\in W}r_S(w)$. The stability bound of the GDMax is $\epsilon$. Therefore, the generalization error of the primal risk is estimated as:
> > $$\zeta_{gen}^{P}(GDMax)\le \sqrt{8 L\ell C_p^2}\sqrt{\epsilon}=8 \lambda \log n \sqrt{\epsilon}.$$
> >
> > On the other hand, $\sum_{i=1}^nz_i/n\in [-\lambda\log n/n,\lambda\log n/n]$
> >  holds with probability at least $1-C/n^{\lambda}$
> >  by Hoeffding inequality.
> > Let $\bar w_S=\epsilon\sqrt{n} (\sum_{z_i\in S}z_i)$.
> >  Then with probability at least $1-C/n^{\lambda}$, $w_S=\bar w_S$
> > Notice that $E_S[r(\bar w_S)-r_S(\bar w_S)]=E_S[\sqrt{n}\cdot (\sqrt{\epsilon} n \sqrt{n})\cdot (\sum_{z_i\in S}z_i/n)^2]=\sqrt{\epsilon}$.
> > It is not hard to show that $|r(\bar w_S)|\le n\sqrt{\epsilon}$,
> >  $r_S(\bar w_S)=0$,
> >  $|r(w_S)|\le 2n\sqrt{\epsilon}$
> >  and $|r_S(w_S)|\le 2n\sqrt{\epsilon}$.
> >
> > Then we have
> > $E_S[r(\bar w_S)-r_S(\bar w_S)]-E_S[r(w_S)-r_S(w_S)]\le 5Cn\sqrt{\epsilon}/n^{\lambda}.$
> >
> > Therefore, $E[r(w_S)-r_S(w_S)]\ge \sqrt{\epsilon}/2 $
> > for sufficiently large $\lambda$ and $n$.
> > Then in this example we have
> > $$\sqrt{\epsilon}/2 \le \zeta_{gen}^P(A)\le 8\lambda\log n \sqrt{\epsilon}.$$
> > For $\epsilon \le 1/n^{\tau+1}$, we have
> >
> > $$\log n\le \frac{1}{\tau+1}\log (1/\epsilon).$$
> >
> > Therefore, the estimate $\zeta_{gen}^P\le \lambda\sqrt{\epsilon}\log(1/\epsilon)/(\tau+1)$
> >
> >  is tight up to a $\log(1/\epsilon)$ factor.
> >
> >
> > We have included this example in the revision (Appendix C.2).
> >
> > **Are there further comparisons between the main results and the previous work?**
> >
> > 1) As stated in the paper, to the best of our knowledge, we are the first to provide generalization bounds in the **nonconvex-concave** regime (without strong concavity). Naively substituting the condition number $\kappa = \infty$ in the bounds for nonconvex-strongly concave cases (from [1][2]) would give us a vacuous upper bound of positive infinity. In this regard, our introduction of $C_e$ and $C_p$ are necessary.
> >
> > 2) In the revised version, we also give the **lower bound** of the generalization error in terms of the stability, which was never considered in the previous papers for studying the generalization for minimax problems. For example, in  [1][2], they did not consider whether the dependence on  $\kappa$ and the Lipschitz constant is optimal.
> >
> > 3) Even for the metric of **primal-dual risk**, our result is the first one that gives the generalization bound for convex-concave problems.
> >
> > 4) Finally, and probably most important, we show that these existing metrics are not sufficient to capture the generalization behavior of minimax learners in certain cases. We thus propose a new metric primal gap, derive its generalization error bound, and even show the tightness of the bound.
> >
> >
> >
> > References:
> >
> > [1] Farnia, F., \&  Ozdaglar, A. (2021, July). Train simultaneously, generalize better: Stability of gradient-based minimax learners. In International Conference on Machine Learning (pp. 3174-3185). PMLR.
> >
> > [2] Lei, Y., Yang, Z., Yang, T., \& Ying, Y. (2021, July). Stability and generalization of stochastic gradient methods for minimax problems. In International Conference on Machine Learning (pp. 6175-6186). PMLR.

---

> > > ### Comment · Reviewer_VSNm · 2022-08-08
> > > **Update after author response**
> > >
> > > I would like to thank the authors for the detailed response to my questions. I am changing my score from 5 to 6 after reviewing the responses.
> > >
> > > One of the reasons I raised the question about dependence on $n$ within $C_p$, $C_e$ and other parameters was that it is unclear whether the dependence on $n$ is captured well in the main theorems. Although the examples provided above in the responses show that in certain special cases, the upper bound is tight up to logarithmic factors, it is unclear whether the same came be said when parameters depend on $n$.
> > >
> > > There is also a philosophical question of whether a large upper bound gives us anything meaningful --- for example, in the author response to reviewer RYz4, the following is quoted: "This quantity should be large for GAN, as discussed in Lemma 5 in our paper. Then, according to Theorem 4, the upper bound of population primal gap is large in this case. This is consistent with our experimental observations and thus justifies our theoretical results." Why is having a large upper bound consistent with experimental observations? Wouldn't a natural upper bound of $\infty$ or naive upper bounds say the same thing? It would seem that a *tight* upper bound with plots of experimental numbers vs the upper bound to show that it is tight would be helpful.
> > >
> > > Taking a look at the other reviews, I agree with reviewer RYz4 that the writing causes some difficulty in reading and the paper can be improved with refined editing in future revisions.

---

> > > > ### Author Response · Authors · 2022-08-09
> > > > **Response to Reviewer VSNm's Update**
> > > >
> > > > First, we would like to thank the reviewer for the open-minded response, and raising the score based on our rebuttal. We would like to further clarify on your concerns as follows.
> > > >
> > > > **$C_e$,$C_p$'s dependence on $n$**
> > > >
> > > > 1.  As discussed above in our response, even if $C_e$ and $C_p$ depend on $n$, the $\log n$ factor does not make major difference in terms of sample complexity, which is the main focus when people are considering generalization in statistical learning.
> > > >
> > > >     For simplicity, we focus on the generalization bound of the [Primal Min Error] $\zeta_{gen}^{PM}$ related to $C_e$ in this response, and the discussion about the primal risk related to $C_p$ is similar.
> > > >
> > > >     The upper bound of the generalization error of [Primal Min Error] is given by $\zeta_{gen}^{PM}\le L_{\theta}^*(C_e)/\sqrt{n}$ and the lower bound given in the example is $L_{\theta}^*(C_e)/(\sqrt{n}\log n)$. The main concern of the reviewer is the case where the upper bound is a **constant** or the lower bound is a **constant** and the reviewer concerns that in each of these two cases the $\log n$ can not be ignored compared to the constant.
> > > >
> > > >     However, this is not an issue when considering the sample complexity needed to achieve certain accuracy. As stated in many papers on generalization, such as [Hardt et al 16], the final goal of studying the generalization bound is to make conclusions on **how many samples people need to achieve a certain small generalization error**. Therefore, specializing to the [Primal Min Error], we need to study how many samples are needed to make $\zeta_{gen}^{PM}\le \delta$ for an arbitrary accuracy $\delta$.
> > > >
> > > >     (a)  If $L_{\theta}^*(C_e)/\sqrt{n}=C_u$ is a constant, then the lower bound given in the example is $C_l/\log n$ where $C_l$ is a constant independent of $n$. Then since the upper bound does not depend on $n$, we can not make $L_{\theta}^*(C_e)/\sqrt{n}\delta$ for small $\delta$ even if $n\rightarrow \infty$. Therefore, the upper bound means that the problem is intractable if $L_{\theta}^*(C_e)/\sqrt{n}$.
> > > >
> > > >      (b)  To make the lower bound $C_l/\log(n)\leq \delta$ in the example, we need $n\geq \exp{C_l/\delta}$, which means an exponential sample size. It is commonly believed in statistical learning that a problem requiring exponential sample size is **intractable**.
> > > >
> > > >     Therefore, we see that this $\log n$ factor does not make major difference (just make changes from infinite sample size to exponential sample size).
> > > >
> > > > 2.  We respectfully disagree that the tightness is just "given in a special case".
> > > >
> > > >     As discussed above in the response, $C_e$ and $C_p$ are problem-dependent and do not necessarily depend on $n$. Then we say our upper bounds are **tight** up to a $\log$ factor because we regard $C_e,C_p$ as constants and we focus on the order of $n$ and $\epsilon$. Also, note that lower bound is all about finding ``special cases'', and is sufficient to show tightness.
> > > >
> > > > **Can upper bound tell us anything?**
> > > >
> > > > 1.  First note that Theorem 4 is **not the only evidence** for explaining the failure of GDMax.  The logic is given below:
> > > >
> > > >     (a)  From Theorem 2, Theorem 4, the upper bound of [Primal Min Error] $\zeta_{gen}^{PM}$ is given by $L_{\theta}^*(C_e)(\Theta_{\theta}^{GDMax})/\sqrt{n}$, which is just equal to $L_{\theta}^*(C_e)/\sqrt{n}$ for GDMAX. As shown by Example 1 and the above discussion, the upper bound is tight even if $L_{\theta}^*(C_e)/\sqrt{n}$ is a constant. Then we claim that $C_e$ is an important constant for studying the generalization of [Primal Min Error], i.e., large $C_e$ can lead to large [Primal Min Error] $\zeta_{gen}^{PM}$ and hence leads to the failure of GDMax.
> > > >
> > > >     (b)  This analysis is further justified by the GAN setting. In this setting, as shown in Lemma 5, we see that for GAN, the constant $C_e\ge \Omega(\sqrt{n})$, which implies that the weight norm of the discriminator attained by GDMax is large, and in Example 2 we see that the [Primal Min Error] is indeed larger than a positive constant. These imply the failure of GDMax.
> > > >
> > > >     Interestingly, these phenomena are also justified by the experiment, which shows that the norm of the discriminator attained by GDMax is indeed large and GDMax can not behave well for GAN.
> > > >
> > > >
> > > >     In conclusion, we give some supporting evidence to show that our upper bounds **are tight**, and with these tight upper bounds, we predict that GDMax can fail for problems with large $C_e$, which is justified by the analytic example and the numerical experiments.

---

> > > > > ### Author Response · Authors · 2022-08-09
> > > > > **Response to Reviewer VSNm's Update (Cont')**
> > > > >
> > > > > 2. Note that good upper bounds of generalization errors are usually used to compare generalization behaviors of different algorithms. In most of the papers about generalization, people just compare upper bounds of different algorithms without proving the tightness, such as the seminal paper [Hardt et al 16] for minimization problems, and the paper [Farnia and Ozdaglar 21] for minimax problems.
> > > > >
> > > > > We believe that our paper provides comparatively more complete analyses, since we also showed **several lower bounds**. Also, our experiments not only justify the final claims but also justify the intermediate conclusions.
> > > > >
> > > > >
> > > > > Thank you very much again for the feedback that help improve the paper.

---

### Official Review · Reviewer_RYz4 · 2022-07-10

**Rating:** 6
**Confidence:** 3
**Soundness:** 3 good
**Presentation:** 2 fair
**Contribution:** 3 good

**Summary:**

This paper studies the algorithm-dependent generalization property of stochastic minimax optimization problems in the nonconvex-concave or nonconvex-nonconcave setting. In particular, this paper gives two (non)concave minimax examples to support that generalization error for primal risk may not be enough to quantify the actual generalization error, due to the objective value differences between population and empirical optimal minimax learners.
To study the generalization error for the primal risk, this paper relies on the tool of $\epsilon$-algorithmic stability [Hardt et al., 2016], and get a bound with sub-linear dependence on the $\epsilon$ for nonconvex-concave problems for the first time. The setting is hard since the Lipschitz continuity and the uniqueness do not hold anymore. Their main idea is to consider a virtual gradient ascent algorithm to fill up this gap.
This paper further investigates the generalization property of GDA and GDmax, and provides two examples and estimate on the capacities to show that GDA generalizes better than GDMax.

**Questions:**

1. The assumption behind Theorem 3 is vague to me. Any references on why the convergence rate on the primal empirical risk can be achieved? The paper by [Du et al., 2019] does not seem to be consistent with the assumption in Theorem 3.
2. Equation (9) has a positive lower bound while Line 238 suggesting that it could attain an upper bound of 0 when $n=1$. Do I miss anything in this example?
3. Some questions about experiments in Appendix D.5. In Line 715, it says "since the stability is improved by using adaptive methods like Adam...". Any references on why this is true? Why not use the vanilla GDA and GDmax? The generated results on MNIST by GDmax are bad and the norm of the discriminator weights seems way too large, (on a scale of 1e7 for an only 4 layers network). Any comment on the parameter tuning in this case?

**Limitations:**

1. The necessity of proposing the "primal gap" can be checked and I think it is an artifact of writing the paper. In Line 220, we see the relation between the generalization error of the primal gap and primal risk is the extra error between the empirical minimax learner and the population one. This term is of course requiring attention since the final target is always to study the primal population risk defined in (2), but it is also algorithm-independent. The analyses of GDA and GDmax still focus on the generalization error of the primal risk. Overall, I think the main contribution should be Lemma 1 and Theorem 7 in the Appendix, and the title might be a little exaggerated.
2. Despite it being the first work establishing generalization bounds for $\epsilon$-stable algorithms without strong concavity, the dependence on $\epsilon$ is not as good as before. While it can be argued that the assumption is weaker, it still draws back the primal population risk and primal dual population risk. For example, in the (non)convex minimization counterpart, such dependence is known to be $\epsilon$ [Hardt et al., 2016]. Furthermore, when the empirical risk is considered together in realization of Assumption 5 to study the population risk, it seems provide worse trade-offs.


**Strengths And Weaknesses:**

Strengths:
1. Solid contribution builds upon recent advances. This paper establishes generalization bounds for stable algorithms in terms of primal risk for (non)convex-concave problems and primal dual risk for convex-concave problems for the first time, relaxing the assumption on the strong concavity or the interchangeability between expectation and maximization.
2. Analytic examples shed light on the understanding the lower bound and the parameter estimation for the generalization of minimax problems.


Weaknesses:
1. The writing can be improved as it causes difficulty even for experienced readers. Examples include but not limit to 1) Last column in Table 1 should refer to Theorem 7 rather than Theorem 6; 2) Using $r$ to denote the risk for minimization problems and primal risk for minimax problem at the same time is confusing; 3) Overall, the paper is very dense and somehow lack of organization, especially Section 4.  For example, Lemma 2 and Proposition 1 can be combined. Lemma 4 is underfull box.

---

> ### Author Response · Authors · 2022-08-02
> **Response to Reviewer RYz4**
>
> We thank the reviewer for the  detailed comments. Please find the response below:
>
> **Writing can be improved:** Thank you very much for the comment regarding writing. We have addressed these comments in the revision (changed $r$ for minimization, combined Lemma 2 and Proposition 1, link Theorem 7 and rewrite Lemma 4).
>
>
> **References for Assumption:**
> In [Du et al 2019], they prove the linear convergence rate of gradient descent using constant stepsize.  In our assumption, we assume that the maximization problems can be solved by gradient ascent in a sublinear $\mathcal{O}(1/t)$ rate  with diminishing stepsize of order $1/t$. The former can usually be used to show  the latter, and this was why we referred this work. Sorry for the small inconsistency.
>
> To address your concern, we have included both cases, with detailed proof and discussions in the revision (see Section 3.5 and Appendix C.3).
>
>
> **Equation (9) has a positive lower bound while Line 238 suggesting that it could attain an upper bound of 0 when n = 1. Do I miss anything in this example?** Thanks for the careful reading. In statistical learning settings we consider  $n>1$.  We have added this condition in the revision.
>
> **Why using Adam?:** We use Adam instead of vanilla GD under the folklore assumption that Adam is more stable for GAN training. See for example [1],  where they analyze Optimistic GD but perform GAN training experiments using Optimistic Adam. Similarly, [2] studies GDA and GDMax, but the experiments on GAN training are done using Adam. In this context, one-step of ADAM descent and one-step of ADAM can be viewed as a GDA and one-step of ADAM descent and many steps of Adam ascent can be viewed as GDMax.
>
> **The generated results on MNIST by GDmax are bad and the norm of the discriminator weights seems way too large:** We note that these two experimental results are exactly what we need. The norm of the discriminator obtained by GDMax, which is $C_e(\Theta_{\theta}^{GDMax})$, is equal to $C_e$ we defined. This quantity should be large for GAN, as discussed in  Lemma 5 in our paper. Then, according to Theorem 4, the upper bound of population primal gap is large in this case. This is consistent with our experimental observations and thus justifies our theoretical results.
>
> **The analyses of GDA and GDMax still focus on the generalization error of the primal risk. Overall, I think the main contribution should be Lemma 1 and Theorem 7 in the Appendix, and the title might be a little exaggerated:** We respectfully disagree that primal gap is  "an artifact of writing the paper", and the main contribution is only Lemma 1 and Theorem 7.
>
> 1) As we show in Example 1, bounding Primal Risk is **NOT** sufficient to conclude generalization behavior of minimax learners. Moreover, based on the response to Reviewer *Z6Bm*, primal-dual gap is **NOT** sufficient, either. Therefore, we propose the new metric: primal gap. The generalization bound of the primal gap is the sum of the two terms, the generalization error of primal risk $\zeta^P_{gen}$ and the Primal-Min Error $\zeta_{gen}^{PM}=E_S[\min_wr_S(w)-\min_wr(w)]$. The second term, Primal-Min Error, when instantiated in Section 4 for the purpose of comparing GDA and GDMax, is **NOT** algorithm-independent. In fact, our Theorem 4 states that for the two different algorithms, their corresponding Primal-Min Error bounds depend on the subset of $\Theta$ that the iterates stay in, which are different for GDA and GDMax.
>
> 2) In Section 4, we not only analyze the generalization error of the primal risk for GDA and GDMax, **but also**  analyzed the Primal-Min Errors of the two algorithms (see Section 4.2).

---

> > ### Author Response · Authors · 2022-08-02
> > **Response to Reviewer RYz4 (cont.)**
> >
> > **$\sqrt{\epsilon}$ seems to be worse than the corresponding bounds for minimization:**
> >
> > Thanks for the careful reading and enlightening comment. Inspired by your comment, we show in the following example that this $\sqrt{\epsilon}$ dependence **is tight**.
> >
> > Consider the following risk function: $f(w,\theta;z)=\sqrt{n/\epsilon}((w/(n\sqrt{n}\epsilon)-z)\theta-\theta^2/(2n\sqrt{n}\epsilon))$,
> > where $w\in W=[-\lambda \epsilon\sqrt{n}\log n,\lambda \epsilon\sqrt{n}\log n]$ and $\theta\in \Theta=\mathbb{R}$.
> > The sample $z$ is drawn from the uniform distribution over {$-1/\sqrt{n},1/\sqrt{n}$}.
> > Then we have $r(w)=\sqrt{n/\epsilon}(w^2/(2\epsilon n\sqrt{n}))$ and $r_S(w)=\sqrt{n/\epsilon}((\epsilon n \sqrt{n})(w/(\epsilon n\sqrt{n})-\sum_{i=1}^nz_i/n)^2/2)$.
> > Now we have $\ell=\frac{\sqrt{n/\epsilon}}{(\epsilon n\sqrt{n})}$, $C_p=\lambda \epsilon\sqrt{n}\log n$ and $L=\frac{\sqrt{n/\epsilon}}{\sqrt{n}}$. If we perform one-step of GDMax with stepsize $1/\ell_{r_S}$ where $\ell_{r_S}=\sqrt{n/\epsilon}/(\epsilon n\sqrt{n})$, then we attain $w_S=\arg\min_{w\in W}r_S(w)$. The stability bound of the GDMax is $\epsilon$. Therefore, the generalization error of the primal risk is estimated as:
> > $$\zeta_{gen}^{P}(GDMax)\le \sqrt{8 L\ell C_p^2}\sqrt{\epsilon}=8 \lambda \log n \sqrt{\epsilon}.$$
> >
> > On the other hand, $\sum_{i=1}^nz_i/n\in [-\lambda\log n/n,\lambda\log n/n]$ holds with probability at least $1-C/n^{\lambda}$
> > by Hoeffding inequality.
> > Let $\bar w_S=\epsilon\sqrt{n} (\sum_{z_i \in S} z_i)$.
> > Then with probability at least $1-C/n^{\lambda}$, $w_S=\bar{w}_S$.
> > Notice that
> >
> > $E_S[r(\bar w_S)-r_S(\bar w_S)]=E_S[\sqrt{n}\cdot (\sqrt{\epsilon} n \sqrt{n})\cdot (\sum_{z_i\in S}z_i/n)^2]=\sqrt{\epsilon}.$
> >
> > It is not hard to show that
> > $|r(\bar{w}_S)|\le n\sqrt{\epsilon}$,
> > $r_S(\bar{w}_S)=0$,
> > $|r(w_S)|\le 2n\sqrt{\epsilon}$
> > and $|r_S(w_S)|\le 2n\sqrt{\epsilon}$.
> >
> > Then we have
> >
> > $$E_S[r(\bar{w}_S)-r_S(\bar{w}_S)]-E_S[r(w_S)-r_S(w_S)]\le 5Cn\sqrt{\epsilon}/n^{\lambda}.$$
> >
> > Therefore, $E[r(w_S)-r_S(w_S)]\ge \sqrt{\epsilon}/2 $
> > for sufficiently large $\lambda$ and $n$.
> > Then in this example we have
> >
> > $$\sqrt{\epsilon}/2 \le \zeta_{gen}^P(A)\le 8\lambda\log n \sqrt{\epsilon}.$$
> >
> > For $\epsilon \le 1/n^{\tau+1}$, we have
> >
> > $$\log n\le \frac{1}{\tau+1}\log (1/\epsilon).$$
> >
> > Therefore, the estimate $\zeta_{gen}^P\le \lambda\sqrt{\epsilon}\log(1/\epsilon)/(\tau+1)$ is tight up to a $\log(1/\epsilon)$ factor.
> >
> > We have included this example in the revision (Appendix C.2).
> >
> > References:
> >
> > [1] Daskalakis, C., Ilyas, A., Syrgkanis, V., \& Zeng, H. (2017). Training GANs with optimism. arXiv preprint arXiv:1711.00141.
> >
> > [2] Farnia, F., \&  Ozdaglar, A. (2021, July). Train simultaneously, generalize better: Stability of gradient-based minimax learners. In International Conference on Machine Learning (pp. 3174-3185). PMLR.

---

> > > ### Comment · Reviewer_RYz4 · 2022-08-09
> > > **Most of my concerns have been addressed**
> > >
> > > I would like to thank the author for addressing my concerns on writing and theoretical results. But I would also like to leave some remark.
> > >
> > > The inconsistency between Assumption 5 and [Du et al., 2019], the discrepancy is not small. Most importantly, [Du et al., 2019] does not consider minimax case and therefore the analysis does not apply to GDA. For GDmax, "The former can usually be used to show the latter" can be more convincing with better reference.
> > >
> > > The complete ineffectiveness of GDmax in experiments still puzzles me despite the norm of the discriminator is large. According to
> > > https://github.com/soumith/ganhacks, a common trick is to train multi-steps on the discriminator rather than on the generator. Consider the opposite but symmetric case: train GAN with ADmin, and the generator this time is a linear function.

---

### Official Review · Reviewer_Z6Bm · 2022-07-11

**Rating:** 5
**Confidence:** 4
**Soundness:** 2 fair
**Presentation:** 3 good
**Contribution:** 2 fair

**Summary:**

The goal of the paper is to understand which metrics (when being small) imply that a solution of the minimax problem (using a dataset) also behaves well on the expected loss. First, the authors provide a negative example showing that “generalization error in primal risk” being small is not sufficient. Then, the authors provide a new metric namely “generalization in the primal gap”, which the authors explain is a better metric. They then consider two popular algorithms for minimax optimization - GDA and GDMax and provide conditions under which one is better than the other.

**Questions:**

Please find my questions in the strength and weakness section above!

**Limitations:**

No foreseeable societal impact.

**Strengths And Weaknesses:**

Strength: The paper considers the important task of understanding the generalization of stochastic mini-max optimization (as an objective function in the primal variable). The problem is well motivated, and has significant potential of impacting practice. My sole reason to suggest a weak acceptance is the relevance of the problem!

Weakness:  I think the paper has several weaknesses:
1. At multiple places, authors mention that primal-dual gap is not sufficient, but I was unable to find a counter-example showing this. Did I miss something?
2. The proof for \zeta^{PG}(A) being small implies a small generalization error is straightforward, but are there cases where $\zeta^{PG}(A)$ is small by  $\zeta^{P}(A)$ is not small? It seems that Lemma 1 and Theorem 2 are bounding $\zeta^{PG}(A)$ by first bounding $\zeta^{P}(A)$, which suggests that $\zeta^{P}(A)$ may very well be the right quantity.
3. The comparison between GDA and GDMax just seems to follow from comparing the upper bounds. Lower bounds are missing, which suggests that the comparison is vacuous.
4. In that sense, are $\zeta^{PG}(A)$ pr $\zeta^{P}(A)$ metrics? Can you compute them a-priori or using data ?

It seems that all that the paper is suggesting is that stable algorithm + additional structural properties, implies that minimax learner generalizes. Is that true? Why are the intermediate quantities $\zeta^{PG}(A)$ pr $\zeta^{P}(A)$ needed?

---

> ### Author Response · Authors · 2022-08-02
> **Response to Reviewer Z6Bm**
>
> We thank the reviewer for the  detailed comments. Please find the response below:
>
> **Is primal-dual gap sufficient?**
>
> No.  In many minimax problems  in  machine learning, due to the ubiquitous nonconvexity, the saddle point does not necessarily exist, see [1][2]. In this case, the primal-dual risk is not a good metric, since it can not be zero,  and the optimality condition in terms of the primal-dual risk is not well-defined. Hence, in the literature, no a priori works used the primal-dual risk for studying minimax problems if there is no saddle-point.
>
> The following example gives a more intuitive explanation about why primal-dual risk is not a good metric in this case.
> Consider the minimax problem:
>
> $\min_{w\in [-100,100]}\max_{\theta\in [-1,1]}-w^2/2+w\theta.$
>
> The minimax solution of this problem is $(w^*,\theta(w^*))=(-100,-1)$ and $(w^*,\theta(w^*)=(100,1)$. The primal-dual risk at the point $(100,1)$ is $200$. On the other hand, consider the point $(99,0)$. The primal-dual risk of $(99,0)$ is $99+199/2=198.5<200$.
> It means that the primal-dual risk of $(99,0)$ is even smaller than the primal-dual risk of the minimax solution $(100,1)$.
>
> However, $(99,0)$ is neither a minimax solution nor a maximin solution nor a saddle point. The main reason for this phenomenon is the absence of saddle points, due to the nonconvexity of the problem. Therefore, primal-dual risk is not a good metric for studying the minimax problems if saddle points do not exist. We are happy include this illustrative example in the paper, if the reviewer advises so.
>
> **Is $\zeta^{P}_{gen}(A)$ the right quantity?**
>
> The goal of Example 1 in our paper is to show that a bound for $\zeta^P_{gen}(A)$ is not enough. In this example, we have $r_S(w) \geq r(w)$, therefore $\zeta^P_{gen}(A) \leq 0$ (which is the best bound we can get!). However, the solution to the empirical problem does not generalize well. At the same time, the generalization error of primal-gap $\zeta_{gen}^{PG}(A)\geq 0.02$, which exactly captures this fact that the solution to the empirical problem does not generalize well. Therefore, this example makes it clear that it is not sufficient to study  $\zeta_{gen}^{P}(A)$ alone, i.e., $\zeta_{gen}^{P}(A)$ being small does not mean good generalization.
>
> The reason behind this is that the second term in $\zeta_{gen}^{PG}(A) $, the [Primal-Min Error] (Equation (8)), is also important in determining if a solution generalizes well. The main message of the paper is that $\zeta_{gen}^{PG}(A)$, which depends on both $\zeta_{gen}^{P}(A)$ as well as [Primal Min Error], overcomes this shortcoming, and correctly encapsulates the generalization behavior of minimax learners.
>
>
> **Comparison of GDA and GDMax are vacuous:**
>
> We note that, we are not only comparing upper bounds. In fact, in Example 3 in the paper,  we establish a lower bound for the generalization error of primal risk, which can be a constant for nonconvex-concave problems. However, for GDA, we can derive an upper bound with $n$ in the denominator.
>
> For the Primal-Min Error  of GDA, we also derive upper bound which is independent of $C_e(\Theta)$, the capacity over the whole set $\Theta$, see  Theorem 4. However, we show in Example 1 that the [Primal-Min Error]  can be a constant for GDMax.
> Therefore, we are not just comparing upper bounds but comparing the upper bound for GDA to the lower bound for GDMax.
>
> **In that sense, are $\zeta^{P}(A)$ and $\zeta^{PG}(A)$ metrics? Can you compute them a-priori or using data?:**
>
> By the arguments above, we believe that both $\zeta^{P}(A)$ and $\zeta^{PG}(A)$ are metrics that can capture the generalization behavior of minimax learners, and our proposal of  $\zeta^{PG}(A)$ is a better one.
>
> No, we cannot compute them a-priori or using data, because one never knows the exact underlying distribution that generates the data to compute them in this learning setting. Such metrics can usually be "estimated/upper-bounded" in terms of algorithmic stability.  This is exactly the framework we are following, similarly as [3].
>
> We remark that, not being able to compute a-priori does not prevent them from being the "metrics", and the same situation happens even in the minimization case, see [3] for example, where the metric cannot be computed either.
>
> **Stable algorithm + Additional structural properties, implies that minimax learner generalizes well. Why do we need $\zeta$?:**
>
> We would like to emphasize that, this statement of "minimax learner generalizes well" cannot be drawn without a proper metric. We attempted to use the same metric as minimization problems, e.g., $\zeta^P(A)$, to capture this. Unfortunately, we **proved** that this does not work. This then motivates us to propose a new metric  $\zeta^{PG}(A)$. With this new metric, we are finally able to conclude the argument above. In this regard, proposing/choosing the right metric is critical to conclude what the reviewer correctly suggested.

---

> > ### Author Response · Authors · 2022-08-02
> > **Response to Reviewer Z6Bm (cont.)**
> >
> >
> > References:
> >
> > [1] Jin, C., Netrapalli, P., \& Jordan, M. (2020, November). What is local optimality in nonconvex-nonconcave minimax optimization?. In International Conference on Machine Learning (pp. 4880-4889) PMLR.
> >
> > [2] Farnia, F., \& Ozdaglar, A. (2020, November). Do GANs always have Nash equilibria?. In International Conference on Machine Learning (pp. 3029-3039) PMLR.
> >
> > [3] Hardt, M., Recht, B., \& Singer, Y. (2016, June). Train faster, generalize better: Stability of stochastic gradient descent. In International Conference on Machine Learning (pp. 1225-1234). PMLR.
> >
> > [4] Farnia, F., \&  Ozdaglar, A. (2021, July). Train simultaneously, generalize better: Stability of gradient-based minimax learners. In International Conference on Machine Learning (pp. 3174-3185). PMLR.
> >
> > [5] Lei, Y., Yang, Z., Yang, T., \& Ying, Y. (2021, July). Stability and generalization of stochastic gradient methods for minimax problems. In International Conference on Machine Learning (pp. 6175-6186). PMLR.

---

### Author Response · Authors · 2022-08-06
**General Response**

Dear Reviewers,

Thank you very much again for your helpful comments, and your time for reviewing our paper. Except for the detailed responses to individual reviewers, we highlight some major changes in our paper and give some common responses here:

1. In the revised version of the paper, we prove that the upper bound of the generalization of primal risk ($\sqrt{\epsilon}$) given in Lemma 1 **is optimal** up to a $\log$ factor.

2. We give more explanations of why primal-gap is a good metric to study generalization in the revised paper. See the comments after the definitions of primal-gap and the discussion in 3.4.

We were wondering is there anything else you would like to discuss? We would very much like to engage with you on our responses to your questions/comments. If you have any remaining questions after reading our response, please do post them here, and we would be happy to discuss further.

Best Regards, the Authors.

---

### Meta-Review · Area_Chair_DYVu · 2022-08-23

**Recommendation:** Accept
**Confidence:** Certain

**Metareview:**

This paper addresses the important problem of deriving generalization for minimax optimization algorithms.  It was shown using concrete examples that the existing measures of generalization (primal risk and the primal-dual risk) may fail to characterize the generalization behavior for minimax problems with nonconvexity.  It then proposed a slightly modified new metric called primal gap. Furthermore, the paper resolved an open question in the literature on how to establish generalization bounds for primal risk and primal-dual risk in the strong sense, i.e., without strong concavity. In the rebuttal, it was shown that the obtained rates are indeed optimal.  All reviewers acknowledged these novel contributions.  I recommend its acceptance accordingly.

As reviewers RYz4 and VSNm mentioned, the presentation and wrting of the paper cause some difficulty in reading the paper and  recognizing the novel contribution of the paper.  I strongly encourage the authors to take their suggestions into account in their revised version.

**Award:**

No

---

### Decision · Program_Chairs · 2022-09-14

Accept